# Shaping Training Load, Technical–Tactical Behaviour, and Well-Being in Football: A Systematic Review

**DOI:** 10.3390/sports13080244

**Published:** 2025-07-25

**Authors:** Pedro Afonso, Pedro Forte, Luís Branquinho, Ricardo Ferraz, Nuno Domingos Garrido, José Eduardo Teixeira

**Affiliations:** 1Biosciences Higher School of Elvas, Polytechnic Institute of Portalegre, 7300-110 Portalegre, Portugal; luisbranquinho@ipportalegre.pt; 2Department of Sport, Exercise and Health Sciences, University of Trás-os-Montes e Alto Douro, 5000-801 Vila Real, Portugal; ndgarrido@gmail.com; 3LiveWell—Research Centre for Active Living and Wellbeing, Instituto Politécnico de Bragança, 5300-253 Bragança, Portugal; pedromiguel.forte@iscedouro.pt; 4Department of Sports, Higher Institute of Educational Sciences of the Douro, 4560-708 Penafiel, Portugal; 5CI-ISCE, ISCE Douro, 4560-547 Penafiel, Portugal; 6Department of Sports Sciences, Instituto Politécnico de Bragança, 5300-252 Bragança, Portugal; 7Life Quality Research Center (LQRC-CIEQV), 2001-964 Santarém, Portugal; 8Department of Sport Sciences, University of Beira Interior, 6201-001 Covilhã, Portugal; ricardo.ferraz@ubi.pt; 9Research Center in Sports Sciences, Health Sciences and Human Development (CIDESD), 5000-801 Vila Real, Portugal; 10Department of Sports Sciences, Polytechnic Institute of Guarda, 6300-559 Guarda, Portugal; 11Department of Sports Sciences, Polytechnic of Cávado and Ave, 4750-810 Guimarães, Portugal; 12SPRINT—Sport Physical Activity and Health Research & Innovation Center, 6300-559 Guarda, Portugal

**Keywords:** monitoring, training load, wellness, player readiness, sub-elite football

## Abstract

Football performance results from the dynamic interaction between physical, tactical, technical, and psychological dimensions—each of which also influences player well-being, recovery, and readiness. However, integrated monitoring approaches remain scarce, particularly in youth and sub-elite contexts. This systematic review screened 341 records from PubMed, Scopus, and Web of Science, with 46 studies meeting the inclusion criteria (*n* = 1763 players; age range: 13.2–28.7 years). Physical external load was reported in 44 studies using GPS-derived metrics such as total distance and high-speed running, while internal load was examined in 36 studies through session-RPE (rate of perceived exertion × duration), heart rate zones, training impulse (TRIMP), and Player Load (PL). A total of 22 studies included well-being indicators capturing fatigue, sleep quality, stress levels, and muscle soreness, through tools such as the Hooper Index (HI), the Total Quality Recovery (TQR) scale, and various Likert-type or composite wellness scores. Tactical behaviours (*n* = 15) were derived from positional tracking systems, while technical performance (*n* = 7) was assessed using metrics like pass accuracy and expected goals, typically obtained from Wyscout^®^ or TRACAB^®^ (a multi-camera optical tracking system). Only five studies employed multivariate models to examine interactions between performance domains or to predict well-being outcomes. Most remained observational, relying on descriptive analyses and examining each domain in isolation. These findings reveal a fragmented approach to player monitoring and a lack of conceptual integration between physical, psychological, tactical, and technical indicators. Future research should prioritise multidimensional, standardised monitoring frameworks that combine contextual, psychophysiological, and performance data to improve applied decision-making and support player health, particularly in sub-elite and youth populations.

## 1. Introduction

Football is one of the world’s most widely practised and studied sports, characterised by a high level of tactical, technical, physical, and psychological complexity [1,2]. The performance of football players results from the interaction between these different dimensions, which do not act in isolation but in an integrated manner, allowing for efficient adaptation to the demands of the game. A football player’s ability to make quick and effective decisions [3], manage fatigue [4], and maintain high performance levels throughout a game or season depends not only on their physical and technical training but also on psychological factors and the monitoring of their well-being [5,6].

Over the last decades, research in football has increasingly focused on optimising performance while ensuring player health and well-being [7,8,9,10], emphasising injury prevention and fatigue management strategies [6,11]. Training and match loads [12] have a direct impact on fatigue and injury predisposition [13,14], making the implementation of effective monitoring and recovery strategies essential [15,16]. Inadequate load management has been shown to compromise athletes’ well-being [9,17], negatively affecting sleep patterns [18], emotional states [4], and recovery capacity [19]. Thus, continuous and multidimensional monitoring of these factors is critical to optimising both performance and health [20].

Several assessment systems have been developed to quantify and analyse the impact of training load [21,22]. Methods such as GPS [23,24] and accelerometers [25,26,27] provide objective data on external loads, allowing for the measurement of variables such as distance covered, number of sprints, accelerations (ACC), and high-intensity efforts [21,28]. Additionally, subjective questionnaires such as the Hooper Index (HI) [29] and the Total Quality Recovery (TQR) scale [30] are widely used to assess players’ perceptions of well-being and physiological responses to training loads [31,32]. However, while these methods effectively identify patterns of fatigue and recovery [33,34], their application in sports practice should be integrated into more holistic approaches, combining objective and subjective measures across multiple performance domains [35].

A key aspect of football performance is the relationship between tactical knowledge and technical skills [36,37], both of which play crucial roles in decision-making and motor efficiency [1,38]. Tactical awareness allows players to perceive, interpret, and respond effectively to dynamic game situations [37], thereby optimising collective play and strategic execution [36]. Meanwhile, the technical proficiency contributes to the quality of tactical and technical actions, and decisions [1] are executed with precision under match pressure [39], facilitating effective passing, dribbling, shooting, and defensive [40]. The Football Tactical Assessment System (FUT-SAT) has been widely used to assess tactical behaviour in football, providing insights into players’ ability to make decisions and perform motor actions based on game principles [41]. By integrating tactical knowledge and technical execution with physical and psychological dimensions, players can optimise their performance [1,42]. However, it remains to ensure the management of fatigue and maintaining well-being [9,17].

Previous systematic reviews have primarily examined performance domains in isolation [43,44], often prioritising physical and psychological variables [43,45,46] while overlooking tactical and technical aspects, as well as their interaction with players’ well-being [5,35]. This fragmented approach may limit the understanding of how training load and recovery strategies influence both athlete health and football-specific performance [2]. Moreover, most of the existing literature has focused predominantly on elite male football players [47,48,49] with limited attention given to sub-elite [12,35,50], semi-professional [51,52], amateur [53,54], or female contexts [55,56]. Yet, players in these categories are subject to different training loads, match structures, recovery resources, and medical support, potentially altering the relationship between performance, health, and well-being [20]. Although youth football deserves consideration due to developmental and maturational aspects [42,57,58], the broader research gap concerns the lack of integrative, multidimensional approaches across the different competitive levels.

Altogether, the present study aimed to conduct a systematic review to examine how the main dimensions of football performance—namely physical, technical, tactical, and psychological—interact with indicators of player health and well-being. This multidimensional perspective seeks to address a critical need in performance science: understanding how combined factors influence recovery, fatigue, and readiness.

While previous research has produced valuable insights into each individual domain, few studies have explored the interdependent nature of these dimensions. This fragmentation may hinder the development of accurate monitoring systems and limit our ability to predict key outcomes such as injury risk, overtraining, or performance decrements.

In this context, we hypothesised that the literature would reveal patterns of association between performance and well-being indicators. Examples include concurrent monitoring of external load (e.g., total distance, high-speed running), internal load (e.g., heart rate zones, TRIMP), technical performance (e.g., pass accuracy), tactical behaviours (e.g., team synchronisation), and well-being status (e.g., HI, TQR, Likert-type scales). Synthesising this evidence is particularly essential in youth and sub-elite populations, where monitoring practices often lack standardisation and integration.

## 2. Materials and Methods

### 2.1. Literature Search Strategy

The search strategy was conducted following the Preferred Reporting Items for Systematic Reviews and Meta-Analyses (PRISMA) guidelines (Appendix A) and followed the Population–Intervention–Comparators–Outcomes–Study Design (PICOS) approach to define the inclusion and exclusion criteria for the studies [59]. The protocol was registered on the International Platform of Registered Systematic Review and Meta-analysis Protocols (INPLASY) under the registration number 202560010 (DOI: 10.37766/inplasy2025.6.0010).

The bibliographic search was carried out in 3 databases—PubMed, Scopus, Web of Science—using keywords and Boolean operators to capture relevant studies on the topic (Table 1) [60,61]. The article collection took place between November and December 2024, ensuring the inclusion of the most recent and relevant publications for the systematic review.

The study selection was carried out independently by the first author (PA) and subsequently verified by a second author (JET). Any discrepancies in the selection of articles were resolved by a third reviewer (PF), following the recommendations of the PRISMA protocol [59]. Only peer-reviewed articles were included, with no prioritisation of authors or journals, ensuring impartiality in the study selection.

Although well-being and psychological outcomes were included as search terms and inclusion criteria, many of the eligible studies merely reported these variables descriptively, without applying integrated or analytical models to examine their relationship with performance indicators.

### 2.2. Selection Criteria

The selection criteria followed the PICOS approach: (1) Population: professional, semi-professional, youth, and sub-elite male and female football players; (2) Intervention: studies that quantified and/or compared variables related to health, well-being, or psychophysiological status (e.g., wellness, readiness, recovery, perceived stress, heart rate variability) and performance (e.g., tactical, technical, physical, or psychological); (3) Comparison: when applicable, periodisation structures (microcycle, mesocycle, or season phase) were compared, although the training organisation itself was not a mandatory inclusion factor; (4) Outcomes: studies that concurrently reported at least one variable related to health, well-being, or psychophysiological status and at least one variable related to performance, regardless of whether a statistical relationship between them was tested; (5) Study Design: experimental and observational studies, including randomised controlled trials (RCTs), cohort studies, and cross-sectional studies.

The inclusion criteria for article selection were as follows: (1) original articles focused on male and female football players from professional, semi-professional, youth, or sub-elite levels; (2) studies that used screening procedures to quantify and/or compare at least one performance-related variable (tactical, technical, physical, or psychological) in relation to internal and/or external training load; (3) studies that concurrently reported at least one variable from the health/well-being/psychophysiological domain and one from the performance domain, even without testing the relationship between them; (4) studies that monitored players for a minimum of one microcycle (≥7 days); (5) studies published from 2014 onward; (6) studies situated within the field of sports performance and sports science; (7) studies with experimental or observational designs (RCTs, quasi-experimental, cohort, cross-sectional); (8) original articles published in peer-reviewed scientific journals; (9) full-text articles available in English or Portuguese; (10) observational studies that achieved a minimum score of 12 out of 16 on the Methodological Quality Checklist for Studies Based on Observational Methodology (MQCOM) [2]; (11) randomised controlled trials (RCTs) that achieved a minimum score of 6 out of 10 on the Physiotherapy Evidence Database (PEDro) scale, to ensure methodological quality [62].

Studies were excluded if (1) they analysed training load in sports other than association football (e.g., Australian football, Gaelic football, or rugby); (2) they included participants under the age of eight; (3) they focused exclusively on biochemical markers or injury intervention protocols; (4) they quantified training load only through field or laboratory tests, without contextual monitoring of internal or external load during training or matches; (5) they monitored players for less than seven consecutive days; (6) they belonged to unrelated research areas or involved non-human participants; (7) they failed to meet minimum methodological quality (MQCOM score < 12 for observational studies or PEDro score < 6 for RCTs); (8) they were not original empirical studies, including systematic reviews, meta-analyses, conference abstracts, opinion papers, commentaries, editorials, books, case studies, non-peer-reviewed publications, master’s theses, or doctoral dissertations.

### 2.3. Quality Assessment

The methodological quality of the included studies was assessed using two validated instruments, according to the study design. For observational studies, the MQCOM was used [63], whereas for intervention studies (randomised controlled and clinical trials), the Physiotherapy Evidence Database (PEDro) scale was applied [64]. This approach is consistent with methodological best practices in systematic reviews involving mixed study designs, where distinct quality assessment tools are applied to ensure design-appropriate evaluation and internal validity. Two independent reviewers (J.E.T. and P.A.) performed the methodological quality assessment. In cases of disagreement, studies were jointly reassessed until a consensus was reached.

The MQCOM checklist [63] is a validated instrument designed to assess research employing observational methodology. It identifies the primary methodological dimensions required for conducting and reporting observational studies and provides a structured quantitative scoring system. The checklist is composed of sixteen items organised into eleven methodological dimensions, including the delimitation of objectives, observational design, specification of participants and observation units, adequacy of observation instruments, software use, data characteristics, specification of parameters, observational sampling, data quality control, data analysis, and interpretation of results. Each item is rated as 0 (does not comply), 0.5 (partially complies), or 1 (fully complies), with items 3 and 15 allowing intermediate scores (0, 0.33, 0.67, or 1). Item 11 may be marked as not applicable when justified. The maximum score achievable on the MQCOM checklist is 16 points, with higher scores indicating better methodological quality.

For intervention studies, methodological quality was assessed using the PEDro scale, which consists of eleven items evaluating key aspects of internal validity and statistical reporting in randomised controlled trials. Each item is scored as 0 (criterion not satisfied) or 1 (criterion satisfied), resulting in a maximum possible score of 11 points. A score of 6 or more is typically considered indicative of high methodological quality. The PEDro scale assesses eligibility criteria specification, random allocation, concealed allocation, baseline comparability, blinding of participants, therapists, and assessors, adequacy of follow-up (≥85% of participants), use of intention-to-treat analysis, reporting of between-group statistical comparisons, and provision of point estimates and variability measures. The PEDro scale has demonstrated moderate-to-excellent inter-rater reliability, with intraclass correlation coefficients (ICCs) ranging from 0.53 to 0.91, and item-level Cohen’s kappa values ranging from 0.36 to 1.00 [64].

### 2.4. Study Coding and Data Extraction

Data extraction of the reviewed articles was organised into the following topics: (1) sampling characteristics by the study design, population, competitive level, sample size (N), sex, age, and expertise level, as presented in Table 2; (2) quality assessment of intervention studies, using the Physiotherapy Evidence Database (PEDro) scale, in which each item is scored (0 or 1) for a total out of 11 points (Table 3); (3) quality assessment of observational studies, based on the Methodological Quality Checklist for Observational Methodology (MQCOM), consisting of 15 individually scored items, providing a total methodological score per study (Table 4).

In cases where specific details such as player age, sex, or competitive level were not reported in the original articles, the corresponding entry in Table 2 was marked as “ND” (Not Described). This reflects incomplete reporting in the source studies and highlights an important limitation in the consistency and transparency of participant descriptions across the reviewed literature.

Data were collected as previously described in the Cochrane Data Extraction Template for Included Studies, and organised using a Microsoft Excel spreadsheet (Microsoft Corporation, Redmond, WA, USA) [65]. The study selection and screening process was conducted using Rayyan software to allow for independent and blinded screening by multiple reviewers [66].
sports-13-00244-t002_Table 2Table 2Summary of the sampling characteristics in the studies included in the systematic review.ReferenceStudy DesignPopulation, Competitive LevelSample (N)SexAge (y)Expertise Level (y)[67]RCTAdult, Professional17Male26.0 ± 2.0ND[68]ObservationalAdult, Elite18MaleNDND[69]RCTYouth, Elite21Male15.3 ± 1.1ND[70]ObservationalAdult, Elite70Male26.6 ± 4.0ND[71]ObservationalYouth, Elite456MaleU15 (*n* = 107) 14.4 ± 0.3
U16 (*n* = 108) 15.4 ± 0.3
U17 (*n* = 104) 16.4 ± 0.4
U19 (*n* = 137) 17.9 ± 0.7ND[72]ObservationalYouth, Elite151MaleU15 (*n* = 56) 14.0 ± 0.2
U17 (*n* = 66) 15.8 ± 0.4
U19 (*n* = 19) 17.8 ± 0.6U15 (*n* = 56) 5.4 ± 1.2
U17 (*n* = 66) 6.8 ± 1.7
U19 (*n* = 19) 9.0 ± 1.7[73]ObservationalAdult, Professional78Male1st DIV (*n* = 32) 24.7 ± 3.8
2nd DIV (*n* = 23) 27.1 ± 3.8
3rd DIV (*n* = 23) 23.1 ± 3.51st DIV (*n* = 32) 6.6 ± 4.3
2nd DIV (*n* = 23) 9.5 ± 3.1
3rd DIV (*n* = 23) 6.6 ± 3.3[74]ObservationalAdult, Professional17Female26.3 ± 4.6ND[75]ObservationalYouth, Academy18Male18.0 ± 1.0ND[76]ObservationalAdult, Elite10Female24.6 ± 2.34.9 ± 2.1[77]ObservationalAdult, Professional17Male23.7 ± 3.26.1 ± 1.6[78] ^1^ObservationalAdult, Professional22MaleNDND[79]ObservationalAdult, Elite30Male28.7 ± 18.68.3 ± 5.7[80]ObservationalYouth, Elite18FemaleNDND[81]ObservationalAdult, College17Female21.8 ± 1.7ND[82]ObservationalAdult, Elite17Male27.8 ± 3.5ND[83]ObservationalAdult, Elite35MaleNDND[84]ObservationalAdult, College19MaleNDND[85]ObservationalAdult, Elite10Male26.0 ± 5.0ND[86] ^2^ObservationalAdult, EliteNDMaleNDND[87]ObservationalAdult, Professional22Male21.7 ± 4.0ND[88]ObservationalAdult, Elite30Male25.0 ± 5.0ND[89]ObservationalAdult, Elite17FemaleNDND[90]RCTYouth, Sub-elite32Male16.1 ± 0.9ND[91]ObservationalYouth, Professional20Male17.4 ± 1.3ND[92]ObservationalYouth, Elite19Male13.3 ± 0.5ND[93]ObservationalAdult, Professional37Male26.4 ± 4.1ND[94]ObservationalAdult, Semi-professional22Female24.6 ± 4.0ND[95]ObservationalAdult, Professional31Male25.4 ± 3.6ND[96]ObservationalAdult, Professional30Male26.0 ± 3.78.5 ± 2.9[97]ObservationalAdult, Elite19Male26.3 ± 4.3ND[98]ObservationalAdult, Professional10Male25.3 ± 2.17.5 ± 2.1[99]ObservationalAdult, Professional30Male24.9 ± 3.17.1 ± 2.8[100]ObservationalAdult, Elite17Male25.4 ± 4.1ND[101]ObservationalAdult, Elite30Male26.2 ± 4.1ND[102]ObservationalAdult, Professional17Male25.0 ± 2.87.4 ± 2.7[103]ObservationalAdult, Professional14Female23.2 ± 5.9ND[104] ^3^ObservationalAdult, Professional22Male24.2 ± 3.5ND[105]ObservationalAdult, Sub-elite28Male20.9 ± 2.4ND[106]ObservationalYouth, Elite15Male18.6 ± 0.4ND[107]RCTYouth, Elite22Male17.2 ± 0.99.7 ± 0.6[108]ObservationalND, Professional42Male27.0 ± 4.0ND[35]ObservationalYouth, Sub-elite60Male15.2 ± 1.8ND[109]ObservationalYouth, Sub-elite60MaleU15 (*n* = 20) 13.3 ± 0.5
U17 (*n* = 20) 15.4 ± 0.5
U19 (*n* = 20) 17.3 ± 0.6U15 (*n* = 20) 4.8 ± 0.9
U17 (*n* = 20) 6.6 ± 1.7
U19 (*n* = 20) 8.8 ± 1.7[12]ObservationalYouth, Sub-elite60MaleU15 (*n* = 20) 13.2 ± 0.5
U17 (*n* = 20) 15.4 ± 0.5
U19 (*n* = 20) 17.4 ± 0.6ND[110]RCTYouth, Sub-elite21 (SAQ group: *n* = 11; SSG group: *n* = 10)MaleSAQ: 9.7 ± 0.4; SSG: 9.5 ± 0.6NDAll Studies--1763-23.9 ± 2.27.3 ± 1.5ND = Not Described; ^1^ 153 matches; ^2^ 128 matches from 3 teams analysed, no player-specific N; ^3^ at least 8 years of expertise.
sports-13-00244-t003_Table 3Table 3Physiotherapy evidence database scale (PEDro) for reviewed intervention groups.ReferenceItem 1Item 2Item 3Item 4Item 5Item 6Item 7Item 8Item 9Item 10Item 11Total Score (Out of 11)[67]110100111118[69]110100011117[90]110100111118[107]110100011117[110]110100111118
sports-13-00244-t004_Table 4Table 4Methodological characteristics observed using the Methodological Quality Checklist for studies based on Observational Methodology (MQCOM).ReferenceItem 1Item 2Item 3Item 4Item 5Item 6Item 7Item 8Item 9Item 10Item 11Item 12Item 13Item 14Item 15Item 16Total[68]011111110.510.51101112.5[70]111111111111111116[71]110.670.51110.50.5111111113.67[72]110.6710.51111111111115.17[73]111111111111111116[74]110.67111111110.5111114.17[75]0.510.670.51110.5010.500.50119.17[76]0.5110.51110.5001110.51112[77]010.670.51110.5010.500.50117.67[78]111111111111111116[79]110.67111111111111115.67[80]110.67111111111111115.67[81]11111110.5110.50.5111114.5[82]0.510.670.51110.50.510.50.510.51111.67[83]0.510.670.51110.5010.500.50119.17[84]0.510.6711110.5011110.51112.67[85]111111111111111116[86]111111110.5111111115.5[87]0.510.6711110.50.5110.510.51112.67[88]111111111111111116[89]0.510.670.51110.5010.500.50119.17[91]110.67111111111111115.67[92]110.67111111111111115.67[93]0.510.670.51110.50.510.500.50119.67[94]110.67111111111111115.67[95]111111111111111116[96]111111111111111116[97]0.510.670.51110.5010.500.50119.17[98]111111111111111116[99]111111111111111116[100]110.67111111111111115.67[101]111111111111111116[102]111111111111111116[103]111111111111111116[104]111111111111111116[105]110.67111111111111115.67[106]111111111111111116[108]110.67111111111111115.67[35]111111111111111116[109]111111111111111116[12]111111111111111116

## 3. Results

### 3.1. Search Results and Study Selection

A total of 341 records were identified through three electronic databases: PubMed (*n* = 121), Scopus (*n* = 73), and Web of Science (*n* = 147). After the removal of 146 duplicate records, 195 records remained and were screened based on titles and abstracts. Of these, 78 full-text articles were assessed for eligibility according to the predefined inclusion and exclusion criteria. Additionally, 5 records were identified through citation searching, and all were assessed for eligibility, with none being excluded at that stage. Following this process, a total of 46 articles were retained for final inclusion in the systematic review—an increase compared to earlier versions of the dataset. Figure 1 presents the PRISMA 2020 flow diagram, outlining the identification, screening, and selection process of the studies included in this review.

### 3.2. Participant Characteristics

A total of 46 studies were included, encompassing a combined sample of 1763 football players. Sample sizes per study ranged from 10 to 456 participants. Of these, 42 studies (91.3%) included male players, and 4 studies (8.7%) included female players [74,76,80,103]. Regarding competitive level, 17 studies (37.0%) involved professional players, followed by 13 studies (28.3%) at the elite level, 12 studies (26.1%) in academy or youth contexts, and 4 studies (8.7%) in sub-elite or semi-professional settings. Concerning the age group, 31 studies (67.4%) focused on adult players, 14 studies (30.4%) focused on youth players (U15–U19), and 1 study (2.2%) included both age groups in a mixed sample. The reported age of participants ranged from 13.2 to 28.7 years. The weighted mean age was 23.9 years (±2.2). Among the studies reporting years of training or competitive experience, the average was 7.3 years (±1.5). Among the study design, 42 studies (91.3%) were observational, and 4 studies (8.7%) were randomised controlled trials.

Table 2 provides a detailed overview of the sample characteristics and study designs included in this systematic review.

### 3.3. Results of Quality Assessment

The methodological quality of the intervention studies was assessed using the PEDro scale. Among the five randomised controlled trials included three studies [67,90,110] that obtained the highest score of 8 out of 11 points, while two studies [69,107] scored 7 points (Table 3). Overall, the methodological quality of the intervention studies was considered good, with consistent fulfilment of criteria such as eligibility specification, random allocation, between-group statistical comparisons, and reporting of variability measures. However, some studies did not meet the criteria related to concealed allocation, participant and assessor blinding, or intention-to-treat analysis. The methodological quality of the observational studies was evaluated using the MQCOM checklist. The scores ranged from 7.67 (lowest quality) to 16.00 (highest quality) out of a maximum of 16 points (Table 4). The mean methodological quality across the included observational studies was 14.34 points, with a standard deviation of 2.56 points. To facilitate interpretation, the observational studies were grouped into categories based on their total scores: Excellent quality (15–16 points): 24 studies (60%); Good quality (13–14.99 points): 8 studies (20%); Moderate quality (10–12.99 points): 6 studies (15%); Poor quality (<10 points): 2 studies (5%). These results indicate that the overall risk of bias was moderate to low across the included studies, although some dimensions—particularly blinding and sampling design—represented recurring sources of potential bias.

### 3.4. Main Findings

#### 3.4.1. Physical Dimensional

Training load monitoring was reported in all 46 included studies, with the external load dimension assessed in 43 studies and internal load in 39 studies. Among external load variables, total distance (TD) was the most frequently reported metric (*n* = 41), typically expressed in metres (m) or metres per minute (m/min). High-speed running (HSR) was assessed in 35 studies, although considerable heterogeneity was found in terminology and thresholds. For example, similar intensity domains were referred to as HSR (≥15 km·h^−1^ [75]), HIR (High-Intensity Running) (≥19.8 km·h^−1^ [68]), or HID (High-Intensity Distance) (>16 km·h^−1^ [74]). Sprinting thresholds also varied, ranging from >5.4 m·s^−1^ to >25.2 km·h^−1^ across studies. A total of 12 studies adopted individualised thresholds based on a percentage of maximum speed (>80% or >85% of max velocity), while the remaining studies used absolute speed cut-offs. ACC and deceleration (DEC) metrics were reported in 33 studies, using varied thresholds from 1.0 m·s^−2^ to >4.0 m·s^−2^. In 9 studies, ACC/DEC were additionally quantified as frequency per minute. Composite indicators derived from triaxial accelerometry were also used, including Player Load (PL) (*n* = 7) and Dynamic Stress Load (DSL) (*n* = 3). Body impacts based on gravitational force zones (g-force) were reported in 3 studies, using classifications from 5.0 to ≥10.1 g. A summary of all measures, thresholds, and metric formulas is provided in Table 5. This heterogeneity in terminology and thresholds—already evident in metrics like HSR—significantly hinders direct comparisons between studies and reinforces the need for greater standardisation in external load monitoring.

#### 3.4.2. Psychological Dimensional

To improve conceptual clarity, psychological monitoring indicators were categorised as either subjective or objective psychophysiological. Subjective tools included the session rating of perceived exertion (sRPE), used in 32 studies and calculated as RPE × session duration (min). Heart rate metrics were reported in 13 studies, including %HRmax (Percentage of Maximum Heart Rate), average heart rate (AvHR), and TRIMP. TRIMP was modelled using the Akubat formula in some cases, and heart rate zones typically ranged from ≤75% to ≥90% HRmax. Affective valence was assessed in 3 studies using the Feeling Scale, while biochemical fatigue markers, such as creatine kinase (CK), were included in 1 [101]. Subjective wellness was assessed in 24 studies. The most common tools were as follows: (1) HI (*n* = 10), based on a 1–7 Likert scale assessing fatigue, sleep quality, stress, and muscle soreness (DOMS); (2) TQR scale (*n* = 6); (3) Well-being Index (WBI) (*n* = 4); (4) Generic Likert-type scales (*n* = 12), addressing perceived recovery, soreness, mood, and stress; (5) composite well-being scores were used in 2 studies, combining multiple psychological variables. Indices reflecting training load distribution and accumulated stress were reported in 20 studies. These included the following: (1) Training/Match ratio (TMr); (2) Training Monotony (TM); (3) Training Strain (TS); (4) Acute: Chronic Workload Ratio (ACWR). In nearly all studies using these indices, weekly loads were derived from cumulative sRPE data.

#### 3.4.3. Technical Dimensional

Technical performance metrics were analysed in 10 studies, focusing on variables such as passes, dribbles, crosses, shots, duels, and goals, most of which were extracted using video analysis software (Wyscout^®^ Spa, Chiavari, Italy, LongoMatch^®^ version 1.5.9, Barcelona, Spain). Metrics were reported either as frequencies or efficiency ratios (e.g., pass accuracy, shot effectiveness). Recent studies have advanced toward contextualised technical analysis, integrating spatial–temporal dynamics, tactical phase, and team strategy. Metrics like expected goals (xG), pass value, and zone-based event frequencies are increasingly being computed using advanced tools such as StatsBomb^®^ and TRACAB^®^, allowing technical actions to be interpreted as part of broader tactical systems.

#### 3.4.4. Tactical Dimensional

Tactical behaviour was assessed in 17 studies, mostly via positional tracking. Reported metrics included the following: (1) team dispersion and contraction indices (*n* = 6); (2) positional entropy (e.g., Approximate Entropy, Shannon) (*n* = 5); (3) synchronisation (longitudinal/lateral) (*n* = 6); (4) dyadic player coordination (*n* = 5); (5) team centroid and positional variability (*n* = 4); (6) ball possession influence on tactical behaviour (*n* = 3). These metrics were often used in conjunction with physical performance indicators. Detailed definitions and operationalisation of tactical constructs are summarised in Table 5, and their methodological applications are outlined in Table 6.

#### 3.4.5. Association Between Dimensions and Player Well-Being

A total of 21 studies analysed more than one performance dimension concurrently, mostly integrating physical and psychological measures such as sRPE, HI, or Heart Rate. However, only two of these studies (9.5%) used advanced statistical techniques to model these associations. Notably, one [109] applied Principal Component Analysis (PCA) to reduce the dimensionality of internal and external training load variables, and another study [105] applied a supervised machine learning model (ordinal regression) to predict players’ well-being index from prior sRPE values. This underutilisation may reflect several barriers, including the need for large, integrated datasets, limited familiarity with multivariate analytics among practitioners, and the absence of standardised frameworks for interpreting outputs in applied settings.

#### 3.4.6. Methodological Approaches

As outlined in Table 6, the included studies employed diverse methodological designs. Studies were predominantly conducted in training contexts (*n* = 36), with some including match-play comparisons (*n* = 26). Game formats varied, though 11 vs. 11 settings were most common, followed by small-sided games (SSGs) and large-sided games (LSGs). GPS technology was the primary tool for external load monitoring (*n* = 35 studies), followed by optical tracking systems (*n* = 5) and local positioning systems (LPM) (*n* = 2). Sampling frequencies ranged from 5 to 25 Hz. Popular GPS systems included Catapult^®^, STATSports^®^, and WIMU PRO^®^, while TRACAB^®^ and Second Spectrum^®^ were frequently used in studies employing video-based tracking. Internal load was typically assessed using sRPE or heart rate telemetry, while some studies incorporated objective field tests (e.g., CMJ, sprint, Yo-Yo IR tests) or biochemical markers (e.g., CK). Psychological well-being was evaluated through validated questionnaires or rating scales. Several studies reported consistent patterns, including (1) progressive tapering on MD-1, (2) underrepresentation of match demands in training sessions, and (3) limited integration of technical/tactical data into holistic monitoring frameworks. These findings underscore the importance of a multidimensional and integrated approach to player monitoring in high-performance football settings.

## 4. Discussion

This systematic review aimed to identify and synthesise the available literature addressing the multiple dimensions of performance in football, particularly in studies that examined physical, tactical, technical, and psychological variables, either independently or in combination. This systematic review provides targeted insights to enhance integrated monitoring practices in football. While methodological advancements are evident, technical and tactical dimensions remain underexplored, and the application of advanced analytics is still limited. The following outcomes highlight practical and research-oriented priorities to address existing gaps and promote the implementation of multidimensional performance frameworks.

Importantly, the fragmentation observed across the literature is not merely methodological—it also appears to stem from structural silos within football science. In many applied and research settings, teams remain compartmentalised, with professionals specialising in distinct domains (e.g., physiology, psychology, performance analysis), often working in isolation. This division is reflected in how data are collected and interpreted, with limited integration between systems, tools, and objectives. For example, GPS and heart rate data may be managed by sport scientists, while tactical metrics are handled by analysts using entirely different frameworks. Moreover, communication between departments is frequently informal or ad hoc, rather than guided by a shared conceptual model. Disciplinary boundaries between quantitative and qualitative traditions further hinder synthesis, especially between physical indicators and contextual tactical data. These structural and epistemological barriers contribute to the observed lack of integration, making it difficult to develop comprehensive monitoring systems that reflect the complex, multidimensional nature of football performance.

Despite the presence of quantitative data across studies, the substantial heterogeneity in outcome definitions, measurement tools, cut-off thresholds, and monitoring durations precluded the implementation of a meaningful meta-analysis. Therefore, a narrative synthesis approach was adopted in accordance with PRISMA recommendations for systematically heterogeneous evidence. The evidence gathered from the 46 included studies reveals a predominant focus on physical load monitoring, while tactical and technical dimensions remain relatively underrepresented. Additionally, although psychophysiological and wellness-related indicators were present in several studies, their integration with other dimensions remains limited. Substantial methodological and contextual heterogeneity—such as variability in age, competition level, load metrics, and outcome definitions—precluded meaningful quantitative aggregation. Rather than being a limitation of the review, this fragmentation reflects the current state of the field and underscores the need for standardised frameworks for future research. These findings underscore a clear gap in the integrated assessment of performance dimensions, particularly within sub-elite football contexts, as revealed through the systematic analysis of the included studies.

### 4.1. Searching Strategy and Coding

This systematic review followed the PRISMA guidelines [59]. For methodological assessment, the Physiotherapy Evidence Database (PEDro) scale was used for randomised controlled trials (RCTs), and the Methodological Quality Checklist for studies based on Observational Methodology (MQCOM) was applied to observational studies. The cut-off values adopted were ≥6 for PEDro and ≥12 for MQCOM, consistent with previous methodological recommendations in sports science literature [2,112]. The searching strategy was aligned with recent systematic reviews in football science [16,113] regarding inclusion and exclusion criteria, database selection (PubMed, Scopus and Web of Science), and keywords relating to training load, performance, periodisation, and player monitoring. The final search outputs yielded 341 articles, from which 78 full texts were assessed for eligibility, culminating in 46 included studies. These procedures ensured methodological consistency and enhanced replicability.

### 4.2. Physical Dimension

Physical performance was the most widely analysed dimension across the included studies. External load was primarily assessed through total distance (TD) [72,74,100,109], distance covered at high-speed running (HSR) [75,76,79,88], sprint thresholds [35,78,98], ACC/DEC [73,74,95,106], and Player Load metrics [76,80,84,94]. The thresholds for HSR and sprinting were relatively consistent, with common benchmarks including ≥19.8 km·h^−1^ for HSR and ≥25.2 km·h^−1^ for sprinting [68,73,78,109]. However, a notable degree of heterogeneity persists across studies. This includes variation in velocity units (e.g., km·h^−1^ vs. m·s^−1^), inconsistent nomenclature (HSR vs. HIR vs. HID), and differences in whether thresholds were absolute or individualised [80,82]. Accelerometry-based measures (ACC/DEC) were employed in 27 studies, typically using multiple intensity zones such as >2 m·s^−2^ or >4 m·s^−2^, which reflect the non-linear movement patterns common in match-play [72,74,76,103]. Composite measures derived from accelerometry, such as PL and DSL, were also applied to quantify cumulative neuromuscular effort [35,76,79].

At the microcycle level, training load analyses revealed consistent tapering patterns throughout the week, with peak intensities typically occurring between MD-3 and MD-4, followed by deliberate reductions in MD-1 [12,75,76,79,102]. However, few studies contextualised these data in relation to tactical demands or match-specific constraints [70,78], limiting the ability to interpret whether training intensity aligns with goals or match-specific positional requirements.

### 4.3. Tactical Dimension

The tactical dimension was represented in 15 of the 46 included studies, primarily through positional data and spatiotemporal analysis. Standard variables included spatial dispersion [72,78,83,109], contraction indexes [70,72], synchronisation measures such as longitudinal and lateral coordination [78,109], dyadic analysis [83,100], centroid dynamics [68,72,86], and adequate playing space [70,78].

Although the adoption of positional tracking systems has increased (TRACAB^®^ [78]; Second Spectrum^®^ [82]; GPS [73,100]), methodological sophistication and standardisation remain heterogeneous (variation in sampling frequency, spatial resolution, and filtering methods). Entropy-based measures (Approximate Entropy, Shannon Entropy) were used in few studies [68,72,109], and even less frequently in combination with physical or technical variables. Stretch indexes were used inconsistently across contexts [72,78], and the application of advanced signal processing methods (e.g., Hilbert transform, multiscale entropy) was absent. Furthermore, few studies examined how tactical organisation interacts with physical or technical outputs [78,83,100], limiting the understanding of the interdependence among performance dimensions. This may be partly explained by the inherent complexity of tactical behaviour, which is fluid, context-dependent, and shaped by situational dynamics such as opponent strategy and match phase. These characteristics make tactical constructs more difficult to objectify and standardise compared to physical or technical measures.

Future investigations should refine the conceptual definitions of tactical constructs, promote standardised data processing, and enhance the integration of tactical data with internal and external load indicators to improve the ecological validity of monitoring practices. Tools like GPS and software such as FUTSAT or TRACAB^®^ could support these developments.

### 4.4. Technical Dimension

Technical actions were analysed in only 7 of the 46 included studies, focusing primarily on event-based metrics such as passes [70,78,86], crosses [86], shots [69,86], dribbles [78,107], and duels [83,86]. These metrics were generally extracted from commercial video analysis platforms such as Wyscout^®^ [86], LongoMatch^®^, or custom-coded match logs [70].

Although these indicators provide useful information regarding individual and collective technical effectiveness, they were rarely contextualised with physical or tactical data. For instance, no study explored how high-speed running or space occupation affects technical execution (for example, pass accuracy under fatigue, or decision-making during high pressing scenarios). The potential for integrated analysis was recognised in studies combining Social Network Analysis with tactical positioning [83] or linking passing behaviour to offensive strategies [78], but such approaches were limited.

Emerging performance models propose contextualising technical actions by field zones, play phases, and pressure situations. However, their practical application remains scarce across the reviewed studies. Future research should prioritise the integration of technical variables with physical and tactical contexts. For example, synchronising video-annotated actions (e.g., passes, dribbles, shots) with GPS-derived external load metrics and internal load indicators such as heart rate (HR) or sRPE could help reveal how technical execution fluctuates under fatigue or positional constraints. Additionally, mixed-method approaches—such as combining spatiotemporal tracking (e.g., TRACAB) with notational analysis, decision-making assessment, or biomechanical markers (e.g., limb kinematics, ground reaction forces)—may offer a more complete understanding of performance under game-like pressure. Controlled small-sided game designs with tactical manipulation could further test these interactions under standardised, yet ecologically valid, conditions.

### 4.5. Psychological Dimension

A total of 36 studies included psychological or psychophysiological measures, with the majority adopting subjective tools such as the sRPE [12,35,76,81,84,87,91,92,109], HI [76,77,97], the TQR [12,35,109], the WBI [87,105], and various Likert-type wellness questionnaires [75,80,84,96].

The sRPE was most frequently applied, typically calculated by multiplying RPE (CR10 scale) by session duration. Despite its widespread use in football research, debate persists regarding its classification—whether it should be treated purely as an internal training load indicator, reflecting physiological strain [114,115], or also as a psychophysiological construct that integrates effort perception with emotional and cognitive feedback [116]. Regardless of classification, its utility lies in its feasibility for daily monitoring and its sensitivity to training-induced fatigue and recovery. However, the use of sRPE and wellness tools was predominantly descriptive. Only a minority of studies examined longitudinal associations between internal load and well-being [75,99,109], limiting predictive insights.

Future work should integrate subjective wellness tools with objective physiological or biomechanical markers to monitor chronic training responses, recovery capacity, and readiness status, particularly in congested schedules or high-risk periods.

### 4.6. Association Between Dimensions and Player Well-Being

A total of 17 studies incorporated more than one performance dimension, yet only a small subset employed true multivariate or longitudinal approaches. Most studies integrated physical and psychological indicators to explore how internal/external loads related to well-being outcomes [75,76,93,98]. These studies indicated that elevated physical load—especially in congested fixtures or periods of poor recovery—was associated with increased perceived fatigue [93,100], impaired sleep quality [87], and greater muscle soreness [77].

However, most designs were cross-sectional or observational in the short term, reducing their predictive capacity regarding overtraining or injury risk. Only a few applied multivariate statistical modelling. Notably, one study [105] implemented an ordinal regression model using machine learning to predict players’ wellness status from prior sRPE values, and another study [109] used PCA to synthesise multidimensional load indicators. Although these methods offer more sophisticated insights into readiness and recovery, they remain largely underutilised in football research.

This limited adoption reflects what may be described as a form of methodological inertia in the field. While advanced modelling techniques such as multivariate analysis and machine learning are increasingly available, their uptake remains limited. This may be due not only to disciplinary conservatism but also to pragmatic constraints, including limited technical training, the steep learning curve associated with complex methods, and the absence of shared frameworks for implementation and interpretation. Furthermore, some people may prioritise simpler tools that are easier to communicate and apply in time-sensitive competitive environments. Recognising these challenges is essential to promoting a gradual shift towards more integrative and analytically robust monitoring practices in football science.

Moreover, interactions between tactical/technical demands and psychophysiological load remain underexplored. Only a limited number of studies [78,83] have attempted to contextualise psychometric data within real game scenarios or cognitive demands, representing an untapped opportunity for more ecologically valid performance analysis.

### 4.7. Study Strenghts, Limitations and Future Directions

Altogether, this systematic review highlights several strengths, including the synthesis of diverse methodologies, performance dimensions, and football contexts. Nevertheless, several limitations must be acknowledged, such as the predominance of cross-sectional study designs, the limited use of predictive or multivariate analytics, the underrepresentation of female and sub-elite cohorts, and the heterogeneity of definitions and measurement protocols across studies.

Future research should aim not only to expand monitoring to neglected populations but also to adopt more integrated and conceptually coherent designs. An ideal multidimensional study would longitudinally track players across physical, technical, tactical, and psychophysiological domains using synchronised data collection systems (e.g., GPS, video tagging, heart rate telemetry, subjective wellness questionnaires). Such a study would incorporate standardised definitions, individualised baselines, and multivariate statistical modelling—preferably through machine learning or latent profile analysis—to detect patterns in training responses and recovery profiles over time [117].

Crucially, the epistemological challenge lies in transcending the traditional segmentation of performance domains. Physical metrics are often treated as objective and quantifiable, while tactical or psychological dimensions are approached qualitatively or subjectively. Overcoming these disciplinary divides will require conceptual frameworks that treat performance as an emergent property of player–environment interactions—dynamic, contextual, and system-based. By advancing toward such integrative, evidence-based monitoring systems, it will be possible to optimise performance while safeguarding health and well-being across all levels of football.

## 5. Conclusions

This systematic review offers a comprehensive synthesis of 46 studies analysing performance and well-being across multiple dimensions (physical, psychological, technical, and tactical). It is among the first to integrate methodological, measurement, and contextual variables across a diverse player sample (youth, adult, elite, sub-elite, and female). Nonetheless, limitations must be acknowledged. These include the predominance of observational and cross-sectional designs, the lack of predictive analytics, underrepresentation of female and sub-elite players, and heterogeneity in definitions and instruments. Future research should prioritise the following: (1) the creation of standardised speed thresholds and acceleration bands to allow comparison of external load across studies; (2) the use of multivariate and machine learning techniques to model how physical and psychological load indicators interact with well-being outcomes over time; (3) the inclusion of tactical and technical variables in longitudinal designs to understand how cognitive demands influence recovery needs; (4) the investigation of training load adaptation in youth and semi-professional players, particularly in settings with limited access to performance technology. Bridging the current disconnect between isolated performance metrics and integrated monitoring will be key to translating data into actionable insights for coaches and sport scientists.

### Key Outcomes

Football performance should move from isolated measures to an integrated approach that combines physical, psychological, technical, and tactical data through interdisciplinary decision-making for coach staffs.Researchers must standardise and justify thresholds for internal load, external load and well-being variables to ensure player data comparability and build robust normative references.Monitoring fatigue, recovery, and readiness should mix subjective and objective measures, including tracking and screening individual trends among young football players.Tactical and technical performance must shift from peripheral to central in monitoring by using specific assessment tools and contextual analysis linked with emotional and cognitive feedback.Future studies should adopt multivariate and ML methods to physical, technical–tactical, and psychological predictors in individual and collective behaviour, classify risk profiles, and generate individualised insights into youth football player development.Practical monitoring must fit training and match conditions by using validated, user-friendly, accessible, and automatized tools, tailoring plans to individual young player needs and fostering player long term development.

## Figures and Tables

**Figure 1 sports-13-00244-f001:**
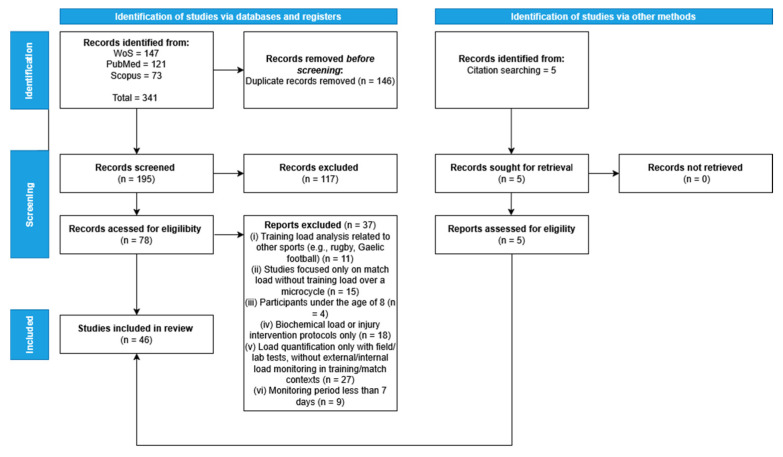
PRISMA flow diagram illustrating the study selection process [111].

**Table 1 sports-13-00244-t001:** Search terms and the following keywords were used in the screening procedures of the systematic review.

Search Item		Keywords
Population	1	“soccer” OR “football” OR “sub-elite players”
Intervention	2	“tactical performance” OR “technical performance” OR “physical performance” OR “psychological performance” OR “external load” OR “internal load” OR “training adaptations” OR “performance quantification”
Comparison/Outcomes	3	Periodisation Structures: “microcycle” OR “mesocycle” OR “season phase” Health and Well-being: “impact on health” OR “well-being” OR “mental health” OR “well-being” OR “health” Load and Performance: “performance indicators” OR “load” OR “load distribution” OR “physiological monitoring” OR “physiological response” OR “physiological adaptations” OR “psychological indicators” OR “tactical demands” OR “technical demands”
Boolean Phrase	4	(Population AND Intervention AND Comparison/Outcomes)

**Table 5 sports-13-00244-t005:** Summary of performance dimensions, measures, measurements, and their thresholds/metric formulas in the included articles.

Dimension	Construct	Measure	Measurement	Description, Thresholds, and/or Metric Formula	Reference
Training load	External Load	Distance and Speed	Speed zones/ Thresholds	HIR: ≥19.8 km h^−1^; Sprint: ≥25.2 km h^−1^	[68]
				Standing Intensity (standing, walking, jogging): <11 km·h^−1^; Low Intensity: 11–13.99 km·h^−1^; Moderate Intensity: 14–16.99 km·h^−1^; High Intensity: 17–20.99 km·h^−1^; Very High Intensity: 21–23.99 km·h^−1^; SPR > 24 km·h^−1^	[70]
				Distance covered per intensity zones: Zone 1: 0–6.9 km·h^−1^; Zone 2: 7.0–9.9 km·h^−1^; Zone 3: 10.0–12.9 km·h^−1^; Zone 4: 13.0–15.9 km·h^−1^; Zone 5: 16.0–17.9 km·h^−1^; Zone 6: ≥18.0 km·h^−1^; Distance covered at sprinting and number of sprints (≥18.0 km·h^−1^)	[72]
				Distance covered (m/min); Distance covered while running: 14.4 km·h^−1^–19.7 km·h^−1^; HSR: >19.8 km·h^−1^; Sprinting: >25.2 km·h^−1^	[73]
				TD (m); RD (m/min); HID: >16.0 km·h^−1^; SPD: >21.0 km·h^−1^	[74]
				TD (m), HSR: ≥15 km·h^−1^, VHSR: ≥19 km·h^−1^	[75]
				TD (m), HSR: ≥15 km·h^−1^, Maximal Speed, Average Speed	[76]
				HIR: 19.8–25.0 km·h^−1^; Sprint: >25.0 km·h^−1^	[78]
				TD (m), HSR: >21 km·h^−1^, HMDL: >25.5 W/kg	[79]
				TD (m), TD/min, HSR: >80% of individual max speed, HSR/min	[80]
				TD (m); TD in HSR (m). HSR > 15 km·h^−1^	[81]
				TD (m); HSR >5.5 m s^−1^; Sprint >7.0 m·s^−1^; MAS (m·s^−1^) = 1200/(time − 20.3); MSS (m·s^−1^) = Maximum recorded sprinting velocity; ASR30 (m·s^−1^) = (0.7 × MAS) + (0.3 × MSS)	[82]
				TD (m); HSD > 5.5 m·s^−1^; Sprints (n) > 25.2 km·h^−1^ (or 7 m·s^−1^)	[88]
				TD (m); HSD > 3.4 m/s; Sprinting (>5.4 m/s)	[89]
				TD (m); LIR < 14.4 km/h; HSR: 19.8–24.98 km/h; Sprint > 24.98 km/h	[91]
				TD (m); HSR: >60% of Individual Max Speed; Sprint > 85% of Individual Max Speed; TD/min (m/min); HMLD (m) > 25.5 W/kg	[94]
				TD/min; HMLD (m) > 25.5 W/kg	[95]
				TD (m), TD/min, SPD (>24 km/h), SPA (n), HMLD (>25.5 W/kg), HMLD/min	[96]
				TD (m); HSD > 19 km/h; Average Speed (m/min)	[97]
				HSR (20–25 km/h); Sprint Distance (>25 km/h)	[98,99]
				TD (m/min); HSR > 19 km/h	[100]
				TD(m); Distance covered per intensity zones: Zone 1: 0–10.9 km/h; Zone 2: 11–13.9 km/h; Zone 3: 14–18.9 km/h; Zone 4: 19–23.9 km/h; Zone 5: >24 km/h	[101]
				TD(m); HSR > 15 km/h	[103]
				TD (m/min); HSD > 6.6 m/s^2^	[106]
				TD (m); HMLD > 25.5 W/kg; SPR > 6.97 m/s; rHSR: 5.5–6.97 m/s; AvS (m/min)	[35]
				TD (m); rHSR 19.8–25.1 km/h; SPR > 25.1 km/h; Sprint count (n); AvS (m/min); MRS (m/s); HMLD > 25.5 W/kg	[12,109]
				TD (m); RD (8–13 km/h); HIR (13–17.9 km/h); SPR (≥18 km/h)	[90]
		Acceleration	Acceleration zones/ thresholds	ACC: 1.0–2.0 m·s^−2^; DEC: >−2.0 m·s^−2^	[68]
				ACC: >3.0 m·s^−2^; DEC: <−3.0 m·s^−2^	[12,35,73,78,109]
				ACC: low (1.0–2.5 m·s^−2^) medium (2.5–4.0 m·s^−2^) high (>4.0 m·s^−2^) total (ACC/min). DEC: low −(1.0–2.5 m·s^−2^) medium −(2.5–4.0 m·s^−2^) high −(>4.0 m·s^−2^)	[74]
				ACC: ≥2.0 m·s^−2^; DEC: ≤−2.0 m·s^−2^	[75,89,91,94,95,98,102]
				ACC: ACC1: >1–2 m·s^−2^ ACC2: >2–3 m·s^−2^ ACC3: >3–4 m·s^−2^ ACC4: >4 m·s^−2^ DEC: DEC1: <−1–2 m·s^−2^ DEC2: <−2–3 m·s^−2^ DEC3: <−3–4 m·s^−2^ DEC4: <−4 m·s^−2^	[76,103]
				ACCHIGH/min; DECHIGH/min; ACC: >3.0 m·s^−2^; DEC: <−3.0 m·s^−2^	[96]
				ACC1: ≥2.0 m·s^−2^; ACC2: ≥4.0 m·s^−2^; DEC1: ≤−2.0 m·s^−2^; DEC2: ≤−4.0 m·s^−2^	[106]
				ACC ≈ 4.0 m/s^2^ (0.5–1.3 s); DEC ≈ 7.0–10.0 m/s^2^ (0.1–0.3 s)	[108]
				ACC (0.5–3.0 m/s^2^); DEC (−0.5 to −3.0 m/s^2^)	[90]
		Accelerometry	Body Impact	Zone 1 (5.0–6.0 g); Zone 2 (6.1–6.5 g); Zone 3 (6.5–7.0 g); Zone 4 (7.1–8.0 g); Zone 5 (8.1–10.0 g); and Zone 6 (≥10.1 g).	[72]
				Count of mechanical body impacts recorded via inertial sensors (e.g., landings, collisions)	[94]
				Number of impacts above 3G measured via inertial sensor	[90]
			Player Load	∑ √((ΔAₓ)^2^ + (ΔAᵧ)^2^ + (ΔA𝓏)^2^)/100, where ΔA = change in acceleration in 3 axes (anteroposterior, mediolateral, vertical)	[76,79,80,84,90,91,94]
			Relative Player Load	Player Load/Duration (min)	[80,84]
			Dynamic Stress Load	Dynamic Stress Load (DSL): derived from triaxial accelerometer (100 Hz); sum of acceleration across 3 axes (X, Y, Z) expressed in arbitrary units (a.u.)	[12,35,109]
	Internal Load	Perceived Exertion	sRPE	RPE × D	[12,35,71,76,77,81,84,87,88,91,92,93,97,98,99,100,101,102,104,105,106,109]
			Differential sRPE	(sRPEbreath × D) + (sRPEleg × D) + (sRPEcog × D)	[80]
			RPE (a.u.)	Borg CR10-scale (0 to 10 arbitrary units)	[12,76,77,79,80,81,84,87,88,91,92,101,102,104,106,107]
				Borg Scale 6–20	[35,67,109]
			NASA-TLX	Mental Workload Score (0–100)	[67]
		Hearth Rate	% HR max	Zone 1 (≤75% HR max), Zone 2 (75–84.9% HR max), Zone 3 (85–89.9% HR max), and Zone 4 (≥90% HR max)	[72]
				%HRmax was analysed as a continuous variable to assess training intensity variations across the season	[88]
				%HRmax = (AvHR/HRmax) × 100	[35,109]
				Zone 1: <80% HR max; Zone 2: 80–90% HR max; Zone 3: >90% HR max	[107]
			HRmax (bpm)	HRmax obtained via Yo-Yo IR1	[35,109]
			AvHR (bpm)	Heart rate monitored via telemetry	[35,67,109]
		Affective Valence	Feeling Scale	A bipolar scale measuring athletes’ emotional responses post-training, ranging from −5 (very bad) to +5 (excellent).	[77]
		Biochemical Marker	Creatine Kinase (CK)	CK values used as marker of fatigue and skeletal muscle damage	[101]
		Training Impulse	Akubat TRIMP	TRIMP = Duration × 0.2053 e^^(3.5179 × HRratio)^	[35,109]
		Neurophysiological Marker	Cerebral Oximetry (NIRS)	Cerebral Oxygenation (StO_2_, O_2_Hb, HHb)	[67]
			Psychomotor Vigilance Test (PVT)	Reaction Time (ms); Lapses of Attention	[67]
		Neurocognitive Marker	Flanker Task	Reaction time (ms); Inhibitory control test: response to target arrow among distractors (congruent vs. incongruent conditions)	[110]
			Visual Search Task	Reaction time (ms); Perceptual speed test: identifying target among distractors; measured at 5-, 10-, 15-, 20-item levels	[110]
	Ratio/scores/tests	Ratio/scores (Weekly TL and ML)	Cumulative Load	∑ x1 + x2 + x3 + x4	[68]
			Training/Match Ratio	TMr= Training Load/Match Load	[98,101]
		Ratio/scores (Weekly TL)	Weekly TL	(sRPE1) + (sRPE2) + … + (sRPEn)	[77,92]
			Weekly Training Monotony	Weekly TL Mean/Weekly TL Standard Deviation	[77,92,100]
			Training Strain	TS = Total weekly load × Training Monotony	[92,100]
			Acute Weekly Load	Total accumulated sRPE in a given week	[92]
			Chronic Weekly Load	Total accumulated sRPE over four weeks	[92]
			Acute: Chronic Workload Ratio (ACWR)	ACWR = Acute Load (1 week)/Chronic Load (4 weeks)	[92,100]
Well-being	Ratios/Scores	Questionnaires	Total Wellness Score	The total wellness score is the sum of the four items. TWS = (Soreness) + (Energy) + (Stress) + (Sleep)	[75]
			Hooper Index	Scale of 1 to 7 for each item: 1 = very low/very good, 7 = very high/very bad. HI = Fatigue + Stress + DOMS + Sleep Quality	[76,77,97]
			Perceived Recovery	Likert Scale (0–10) 0 = Extremely poor recovery/extreme fatigue; 5 = Adequate recovery; 10 = Excellent recovery/high energy levels	[80]
			Perceived Well-being	Likert Scale (1–5) 1 = Always tired/insomnia/very sore/very stressed/very irritable/depressed; 5 = Very refreshed/very rested/very well/very relaxed/very positive	[80,87,98,99,102]
				Likert Scale of 1 to 7 for each item (1 = very good/very low, 7 = very poor/very high). Total score = (Fatigue) + (Sleep Quality) + (Stress) + (Muscle Soreness)	[95]
			Lower Extremity Soreness	Likert Scale (0–6) 0 = No pain; 1 = Light pain (only when touched); 2 = Moderate pain (slight persistent pain); 3 = Light pain (stairs); 4 = Light pain (walking flat); 5 = Moderate pain (stiffness, weakness); 6 = Severe pain (limits movement).	[84]
			Well-being Index	WBI = Fatigue + Sleep Quality + DOMS + Stress + Mood. Higher scores indicate better well-being.	[87]
				Likert scale 1–5 for each item (fatigue, sleep, pain, stress, mood); 1 = Optimal well-being, 5 = Poor well-being; WBI = Sum of all 5 items (higher = worse)	[105]
			Perceived Muscle Soreness and Fatigue	Borg CR-10 Scale (1–10) 1 = minimal soreness/fatigue, 10 = maximum soreness/fatigue	[89]
			Total Perceived Response Score	Composite Score (1–7 per item × 5); Fatigue, muscle soreness, psychological status, sleep quality, and sleep duration. 1 = very, very good; 7 = very, very poor.	[93]
			Total Quality Recovery	Total Quality Recovery scale: 6 (very, very poor recovery) to 20 (very, very good recovery)	[12,35,109]
		Records	Sleep Hours	Hours (h): Players recorded total hours slept the previous night	[89]
			MVPA, Steps/day	MVPA = time in moderate (757–1111 cpm) + vigorous (≥1112 cpm); steps/day also tracked	[90]
			Sedentary Time	Sedentary = 0–180 counts/min; SB bout = ≥30 min uninterrupted	[90]
Expertise	Technical Variables	Individual Actions	Passes, Crosses, Dribbles, Duels, Shots	Ball possession (s), passes (n), passing accuracy (%), 1-on-1 duels (n), wins in 1-on-1 duels (n), and wins in 1-on-1 duels (%)	[70]
				Dribble success (%): Player retains possession and surpasses an opponent Pass types (n): Short (<10 m), Medium (10–30 m), Long (>30 m). Success = reaching a teammate	[78]
				Goals (n); Shots (n); Shots on target (n); Crosses (n); Accurate crosses (n); Offensive duels (n); Offensive duels won (n); Effectiveness (%) = Shots on target × 100/Shots	[86]
				Passes (n); Dribbles (n); Goals (n); Game Pauses (n)	[107]
			Ball Speed	Ball velocity measured with radar after a kick from 11 m	[69]
		Performance	Networks	Flow Centrality (CFC): Measures a player’s involvement in team passing sequences = ∑[k = 1 to m] pk(ni)/M Flow Betweenness (CFB): Identifies a player’s role as a “bridging” passer between teammates = CFB(ni) = ∑[k = 1 to m] bk(ni)/M Weighted Betweenness (CWB): Evaluates playmaking importance based on the strongest passing links between teammates = ∑[j ≠ k ≠ i] (gi_jk/gjk)	[83]
	Tactical Variables	Ball Possession	Offensive/Defensive Phases	Index of Game Control (IGC = zPA + zPD + zTP + zPS + zPSF)	[78]
				Successful possession: Team enters finishing zone (proxy for goal scoring). Unsuccessful possession: Ball lost before entering the finishing zone. Neutral possession: Starts in the finishing zone or enters via set-play.	[83]
				Ball possession (%): The time when a team takes over the ball from the opposing team without any clear interruption, as a proportion of the total time the ball was in play	[86]
		Style of Play	Offensive/Defensive Phases	Index of Offensive Behaviour (IOB = IGC + zRP + zDPA + zGP-zTA + zGS + zMPA)	[78]
	Measuares	Tests	Sprint Test	Repeated Sprint Ability: 10 × 20 m	[67]
			Counter Movement Jump	h = (f^2^ × g)/8	[75]
				Optojump Photoelectric System	[87]
				Measured using Chronojump	[106]
			Aerobic Power Test (30-15 IFT)	VO2_max_((mL/kg/min)) = 28.3 − (2.15 × 1) − (0.741 × age) − (0.0357 × weight) + (0.058 × age × VIFT) + (1.03 × VIFT)	[87]
			Yo-Yo IR Test Level 2	Total distance covered in shuttle runs (20 m) with increasing intensity until failure	[89]
			Yo-Yo IR1 Test	VO_2max_ Estimated using total distance covered in the Yo-Yo Intermittent Recovery Test Level 1	[106]
			30 m Sprint Time (s)	Best of 3 attempts; standing start; measured using photocell timing system	[106]
			Illinois Agility Test	Best of 3 attempts; standard IAGT setup with cones and timing gates	[106]
			1 Rep Max	1-RM Bench Press (kg)	[106]
				1-RM Squat (kg)	
			Squat Jump	Measured using Chronojump	[106]
			5 m Sprint Time (s)	Linear sprint over 5 m; best of 3 trials recorded with timing gates	[69,110]
			15 m Sprint Time (s)	Linear sprint over 15 m	[69]
			20 m Sprint Time (s)	Linear sprint over 20 m; best of 3 trials recorded with timing gates	[110]
			Change of Direction (COD90)	Timed zigzag sprint with five 90° turns; best of 3 trials	[110]
			Complex Contrast Training (CCT)	Combination of strength exercises (80–90% 1RM) with explosive tasks (sprint, jump, header)	[69]
			505 Agility Test	Classic test involving 180° turn around a cone; measures COD ability	[69]
		Anthropometric Measures	BMI (kg/m^2^)	BMI = Weight (kg)/Height (m)^2^	[87]
			Body Composition	Body fat (%); Lean Mass (%); Total Body Mass (kg)	[89,106]

**Table 6 sports-13-00244-t006:** Methodological approaches of included articles.

Reference (Year)	Study Purpose	Experimental Approach						Methodological Procedures	Data Collection (Device Specification)	Main Findings	Key Outcomes
		Match-Play	Training Set	Game Format	Physical/ Physiological	Positional/ Tactical	Other Dimensions				
[67]	Examine the effect of mental fatigue on repeated sprint ability and psychomotor vigilance.	✗	✓	10 × 20 m sprints	✓	✗	✓ (Technical)	RSA test post-Stroop vs. control; cognitive and physical performance tests	Polar HR monitor, PortaLite NIRS, RPE (6–20), PVT	Mental fatigue reduced RSA performance; NIRS and PVT responses altered post-Stroop.	Cognitive fatigue impairs anaerobic performance; mental state affects physical output.
[68]	Quantify and compare the most demanding 5 min passages of play and the accumulated training load relative to match demands per playing position.	✓	✓	11 vs. 11 (full match)	✓	✓	✓ (Psychological)	GPS (ZXY, Hz not reported): TD (m), HIR (m), Sprint (m), ACC/DEC (n); 5 min peak analysis; position-specific comparison	ZXY Sport Tracking System (radio-based local positioning); Measures: accelerations, decelerations, HIR, sprints, 5 min peaks	Match demands overperformed for acc/decc; underperformed for HIR/sprints; WB showed lower sprint peaks in training.	Training underperformed sprint and HIR vs. match; overperformed acc/dec; positional mismatches in training demands.
[69]	Examine effect of CCT frequency on sprint, jump, agility and kick performance in youth football.	✗	✓	Integrated drills (team-based)	✓	✗	✗	6-week intervention: 2× vs. 3× per week vs. control; performance testing pre-post	Radar (kick speed), jump mat (CMJ, SJ), 505 test, sprint timing system	3×/week CCT improved sprint and kick speed more than 2×/week or control.	Training frequency impacts physical metrics; higher frequency improves sprint and kick metrics.
[70]	Investigate physical and technical performance variations across six phases of three Bundesliga seasons.	✓	✗	11 vs. 11 (official Bundesliga matches)	✓	✓	✓ (Technical)	FUTSTAT (VIS.TRACK, 25 Hz): TD (km), speed zone distances, passes, ball possession; seasonal phase comparison	VIS.TRACK vision-based tracking system (25 Hz); data from 918 Bundesliga matches	Performance decreased in later season phases; peak in physical efforts during phase 4; stability in technical actions.	Physical performance declined after 2/3 of season; stability in technical actions; peak in phase 4.
[71]	Describe weekly microcycle training load distribution across age groups (U15–U19) in elite youth soccer.	✓	✓	Standard academy microcycle (MD-4 to MD; 11 vs. 11, age-specific sessions)	✓	✗	✓ (Psychological)	RPE (CR10); session duration (min); training load as sRPE (AU); analysis by age group (U15–U19) and weekly day (MD-4 to MD)	Session-RPE (CR10 scale), session duration (min), calculated sRPE-load; standardised microcycle structure (MD-4 to MD)	Match day had highest training load; U19 had lower durations midweek; training load progression with age group; reduced load MD-1.	Match day has highest load; older groups train longer; tapering evident in MD-1.
[72]	Describe weekly time–motion and physiological demands in U15, U17, and U19 elite Portuguese football players.	✗	✓	Post-, mid- and pre-match sessions: SSG, analytical drills; U15–U19	✓	✓	✗	GPS (15 Hz, GPSports): TD (m), sprints (>18 km/h), impacts (g); HR (Polar): HRmax zones; analysis by week moment (post, mid, pre-match)	15 Hz GPS (SPI Pro, GPSports); Polar HR monitors; HR zones, distance in speed zones, accelerations, impacts, sprints	Younger players had higher sprint and high-intensity efforts midweek; pre-match loads decreased with age; physical demands varied with age and session.	U15 showed higher sprint/high speed midweek; U19 reduced pre-match load; load varies by age and session.
[73]	Compare microcycle external load distribution across three competitive levels (1st, 2nd, 3rd divisions) in Portuguese football.	✓	✓	GK + 10 vs. 10 + GK; SSG, LSG, tactical, and technical exercises	✓	✓	✗	GPS (10 Hz, Catapult Vector S7): TD (m/min), HSR (>19.8 km/h), sprint (>25.2 km/h), ACC/DEC (>3 m/s^2^); comparison across competitive levels (1st, 2nd, 3rd DIV)	10 Hz GPS (Catapult Vector S7); variables: total distance, HSR, sprints, accelerations/decelerations, per minute of play	1st DIV had higher volume overall; 2nd DIV emphasised MD-2 load; 3rd DIV showed more accelerations; MD-1 had the lowest load for all.	1st DIV covered more volume; 3rd DIV had more accelerations; MD-1 consistently exhibited lowest load.
[74]	Analyse inter- and intra-microcycle external load in female professional players by position.	✓	✓	GK + 10 vs. 10 + GK; MD-4 to MD; Spanish 1st Division (women)	✓	✓	✗	GPS (10 Hz) + Accelerometer (100 Hz, WIMU PRO): TD, HSR, Sprint (m), ACC/DEC (n), Max Speed (km/h), PlayerLoad (AU); positional and sessional comparison	10 Hz GPS + 100 Hz accelerometer (WIMU PRO, RealTrack Systems); external load: TD, HID, sprints, ACC, DEC, max speed, PlayerLoad	MD-3 had highest load; match had highest HID and sprinting; FWs showed more sprints than CMs and CBs; variability in MD-2.	FWs sprinted more on MD and MD-2; MD-3 had highest load; significant inter- and intra-day variation
[75]	Explore if CMJ and wellness scores detect postmatch fatigue and predict subsequent match physical output in elite youth soccer.	✓	✓	11 vs. 11 (U18 competitive matches)	✓	✗	✓ (Psychological)	GPS (10 Hz) + Accelerometer (200 Hz, Polar): TD, HSR, VHSR, ACC/DEC; CMJ (jump mat, height in cm); wellness questionnaire (sleep, stress, energy, muscle soreness)	10 Hz GPS + 200 Hz accelerometer (Polar Team System); Wellness (sleep, energy, stress, soreness); CMJ via jump mat	CMJ and wellness scores showed post-match fatigue effects; wellness at MD-5 predicted next match’s acceleration/deceleration output.	Wellness scores at MD-5 predicted match acceleration/deceleration; CMJ and wellness sensitive to post-match fatigue
[76]	Monitor changes in wellness (sleep, stress, fatigue, soreness) and affective valence during pre-season vs. in-season in pro soccer players.	✓	✓	Technical, tactical, and physical sessions: PT, TT, PTT (full squad)	✓	✗	✓ (Psychological)	RPE (CR10): Load (AU), Monotony, Strain; HI (sleep, stress, fatigue, DOMS); Feeling Scale (FS, −5 to +5); pre-season vs. in-season comparison	Subjective ratings (Hooper Index: sleep, stress, fatigue, soreness); RPE-based load (session-RPE); FS (affective valence)	Higher fatigue, stress, soreness, and load in pre-season; lower affective valence vs. in-season; technical sessions induced better feeling.	Pre-season showed higher fatigue, stress, soreness and lower affective valence vs. in-season
[77]	Quantify external, internal load and wellness markers across a typical in-season microcycle in elite women’s football.	✓	✓	11 vs. 11 (MD-5, MD-4, MD-2 sessions)	✓	✗	✓ (Psychological)	GPS (10 Hz, PlayerTek): TD (m), HSR (>15 km/h), Sprint (m), ACC/DEC (m/s^2^), Max Speed; RPE (CR10); Hooper Index (sleep, stress, fatigue, DOMS)	Data collected using GPS, RPE and Hooper Index; sessions MD-5, MD-4, MD-2, match	Training and matches (RPE, HI, GPS); measures collected daily across week.	Match had highest external intensity; MD-2 lowest; wellness stable across microcycle.
[78]	Investigate how offensive playing style (ball possession vs. counter-attacking) affects physical, technical, and success variables.	✓	✗	11 vs. 11 (Bundesliga matches)	✓	✓	✓ (Technical)	TRACAB optical tracking system: ACC/DEC (>3 m/s^2^), HSR (19.8–25 km/h), Sprint (>25 km/h); event data: passes, dribbles, success rate; PSC model	Tracking (TRACAB) and event data; data extracted during ball possession phases	Tracking and event data analysed per team during possession phases.	Ball possession teams performed more ACC/DEC; counter-attacking ran more per second in possession
[79]	Analyse training load distribution and well-being in elite youth soccer players over a season.	✗	✓	Typical training microcycles (U15-U19)	✓	✗	✓ (Psychological)	RPE (CR10), HI (fatigue, soreness, stress, sleep), session duration; CMJ measured weekly.	Subjective questionnaires; CMJ test with contact mat	Variation in wellness and CMJ across microcycles; younger players with higher loads.	Younger players showed higher weekly load; HI and CMJ varied across weeks; need for individualised monitoring.
[80]	Examine match demands and acceleration profiles in different youth age categories.	✓	✗	11 vs. 11 (official youth matches, U15–U19)	✓	✓	✗	GPS: TD (m), ACC/DEC (n), Max Speed (km/h), HSR (>18 km/h), PlayerLoad; analysis by position and age	GPS (10 Hz, WIMU PRO); Accelerometer (100 Hz)	Acceleration/sprint demands vary with age; match demand progression from U15 to U19.	U15/U17 had higher ACC/DEC, while U19 had more HSR; match demands increase with age.
[81]	Analyse how competition phase and position affect the relationship between internal and external load in female collegiate soccer.	✓	✗	11 vs. 11 (NCAA D1 women’s soccer)	✓	✓	✓ (Psychological)	ETL: GPS (10 Hz): TD (m), HSR (>15 km/h); ITL: sRPE (CR10 × min)	Catapult OptimEye S5 (10 Hz); RPE form (Google Forms)	Internal load affected by total distance, not HSR; forward players report higher sRPE.	TD and position (forwards) predicted sRPE; HSR was not significant; stronger dose–response by role.
[82]	Analyse positional distances covered above generic and individualised speed thresholds during most demanding periods.	✓	✗	11 vs. 11 (EPL, official matches)	✓	✓	✗	Optical tracking: TD (m), HSR (>5.5 m/s), Sprint (>7 m/s), MAS, MSS, ASR30; rolling averages (1–10 min)	Second Spectrum (25 Hz); MAS via shuttle test (1200 m)	Individualised thresholds more precise than generic ones; defenders have lower MAS and sprinting.	Individualised metrics showed better precision than generic ones; central defenders had lowest MAS/HSR.
[83]	Identify dominant and intermediary players via play-by-play social network analysis.	✓	✗	11 vs. 11 (Bundesliga, 70 matches)	✗	✓	✓ (Technical)	SNA metrics: flow centrality, flow betweenness, weighted betweenness; positional impact per phase	TRACAB multi-camera (25 Hz), Bundesliga DFL database	Midfielders key in successful plays; defenders central in general, but not decisive.	Central defenders dominant in general plays; midfielders more involved in successful plays.
[84]	Analyse relationship between external loads, sRPE-load, and perceived soreness across a season.	✓	✓	11 vs. 11; training and matches, NCAA DIII	✓	✗	✓ (Psychological)	GPS (10 Hz): TD (m), Sprint (m), ACC/DEC; RPE (CR10); soreness (Likert 0–6); PlayerLoad	Catapult PlayerTek (GPS), CR10 scale, soreness scale	sRPE strongly correlated with GPS variables; soreness only weakly related to load.	sRPE strongly correlated with GPS variables; soreness weakly correlated with load.
[85]	Predict HR responses to training drills using GPS metrics; use delta HR as fitness indicator.	✗	✓	SSGs (5 vs. 5 to 10 vs. 10), elite team (PSG)	✓	✗	✓ (Psychological)	GPS (5 Hz) + Acc (100 Hz): TD, HS, VHS, force load, mechanical work; HR monitored during drills	SPI-Pro GPSports + Polar H1 (synchronised); ADI software	HRΔ provides useful daily fitness marker; correlated with submaximal HR and seasonal trends.	Predicted vs. actual HR differences tracked fitness changes; HRΔ decreased with fitness improvements.
[86]	Assess changes in technical indicators over match time and between UCL stages and locations.	✓	✗	11 vs. 11 (UEFA Champions League, 128 matches)	✗	✓	✓ (Technical)	Wyscout stats: shots, shots on target, crosses, possession, offensive duels, etc., per 15 min segments	Wyscout platform (match event data)	Technical variables increase in final 15 mins; match stage and location influence performance.	More offensive actions in 2nd half; home teams and group stage showed higher technical indicators.
[87]	Monitor ITI, well-being, and CMJ over 5-week pre-season in Croatian professional players.	✗	✓	Technical-tactical and strength-conditioning sessions	✓	✗	✓ (Psychological)	sRPE (CR10); HI (fatigue, DOMS, sleep, mood, stress), CMJ (Optojump), monotony, strain	CR10 RPE scale; HI questionnaire; Optojump CMJ	Fatigue, DOMS and WBI correlate with ITI; CMJ improves over 5 weeks of training.	ITI increased over weeks; CMJ improved; well-being correlated negatively with strain/load.
[88]	Quantify seasonal training load in elite EPL players across pre- and in-season phases.	✓	✓	11 vs. 11; full team sessions (pre- and in-season)	✓	✓	✗	GPS (5 Hz): TD, HSR; HR (%max); RPE × duration; analysed by week, mesocycle, and MD.	GPSports SPI Pro X; HR telemetry; Borg CR10	Training load lower on MD-1; consistent loads MD-2 to MD-5; stable across season.	Training load reduced on MD-1 only; consistent load for MD-2 to MD-5; limited variation by mesocycle.
[89]	Investigate running activity during male professional soccer matches at different competition levels.	✓	✗	11 vs. 11 (professional match analysis)	✓	✓	✗	GPS (10 Hz): TD (m), HSR (>18 km/h), Sprint (>23 km/h), PlayerLoad (AU)	GPS (MinimaxX Team 4.0, Catapult Innovations, Australia); 10 Hz, match-day data	Higher-level teams covered more distance in high-speed and sprint zones than lower-level teams.	Sprint and HSR differentiate match demands across levels; tactical context influences load distribution.
[90]	Test effectiveness of wearable wristbands (REM vs. nREM) on PA/SB and training responses	✗	✓	11 vs. 11 training sessions	✓	✗	✗	Monitoring over 2 weeks with and without PA reminders; training monitored with WIMU	ActiGraph GT9X (30 Hz), WIMU PRO (GPS + IMU), Fitbit Charge 2	No significant differences in PA or training load between REM and nREM; wearable reminders had no effect.	Wearables alone do not change PA or SB; ineffective for optimising training responses in youth players.
[91]	Evaluate the validity of RPE as a tool to monitor training intensity/load in elite football players.	✗	✓	Full-team training sessions	✓	✗	✓ (Psychological)	Session-RPE (CR10); correlation with HR zones and subjective fatigue; training duration	Subjective RPE scale; Polar HR monitors (real-time HR data collection)	RPE highly correlated with internal load; useful for non-invasive monitoring.	RPE valid across training types; strong correlation with HR-based measures of internal load.
[92]	Assess training monotony, strain and ACWR across 4 weeks of in-season microcycles in pro players.	✗	✓	Technical-tactical and physical training sessions	✓	✓	✓ (Psychological)	s-RPE; Total Distance (m), HSR (>19.8 km/h); computed Monotony, Strain, ACWR	RPE (CR10); GPS (10 Hz, WIMU PRO); monitoring during full microcycles	High week-to-week variability in ACWR and training monotony; peaks correspond to match congestion.	Monitoring workload variation is key to managing training load and minimising injury risk.
[93]	Compare perceived fatigue/recovery between congested vs. non-congested microcycles in national team tournaments.	✓	✓	11 vs. 11 (official international matches)	✓	✗	✓ (Psychological)	s-RPE × duration (AU); HI (fatigue, soreness, stress, sleep quality); subjective response score	CR10 RPE; wellness questionnaires; SMARTABASE app	Congested schedules impaired pre-match perceived recovery; no worsening of post-match fatigue.	Player-reported wellness tools useful to track readiness; congestion affects pre-match status more than post-match recovery.
[94]	Compare external load of training sessions vs. official matches in elite female players.	✓	✓	11 vs. 11; OM, INT, EXT, VEL, PREOM sessions	✓	✓	✗	GPS: TD, HSR, Sprint, ACC/DEC, PL, TD/min, HMLD, Impacts	WIMU PRO (10 Hz), RealTrack Systems; SPro software; individual session tracking	Training loads varied by session type, but always lower than match; PREOM exhibited lowest intensity.	Different session types target distinct capacities; none fully replicated match demands.
[95]	Examine how microcycle length affects external load and perceived wellness in LaLiga Smartbank players.	✓	✓	11 vs. 11; short, regular, long microcycles	✓	✗	✓ (Psychological)	GPS: Distance (m), ACCHIGH, DECHIGH, Sprinting actions/dist, HMLD; Wellness questionnaire	WIMU PRO (RealTrack Systems); EPTS-certified GPS; daily load and wellness tracking	Microcycle length impacts training volume and intensity, but not wellness scores.	External load varies with microcycle duration; wellness stable despite training variability.
[96]	Identify key load indicators and variability across training/match days via PCA in elite soccer.	✓	✓	11 vs. 11; full-season data by training day (MD±)	✓	✓	✗	111 GPS variables reduced by PCA: TD, HSR, acceleration zones, FFT duration, metabolic power	WIMU PRO (RealTrack); PCA for dimensionality reduction and day-wise variability	MD-1 had lowest load variability; MD + 1 most variable; 7 indicators explained 80% variance.	Load programming shows systematic tapering; PCA aids in simplifying monitoring strategy.
[97]	Quantify TL in one-, two-, three-game week microcycles in elite UEFA-level teams.	✓	✓	11 vs. 11; M1–M5 microcycles	✓	✓	✓ (Psychological)	GPS: TD, Speed zones (5), s-RPE, CK (fatigue marker)	GPS (10 Hz, STATSports); RPE (CR10); CK via blood sample	Training volume and CK increased with match frequency; external load tapered MD-1.	More matches = more load; MD-1 shows clear tapering pattern regardless of game count.
[98]	Report in-season internal/external TL variation across 10 mesocycles in UEFA team.	✓	✓	11 vs. 11 (training and match days)	✓	✓	✓ (Psychological)	s-RPE (CR10); GPS: TD, HSD, Av Speed; Hooper Index: fatigue, stress, sleep, DOMS	Viper GPS (10 Hz, STATSports); subjective HI daily monitoring	Stable internal/external TL across season; MD-1 had lowest load; positional differences were minimal.	Typical microcycle load progression: early-week peak, MD-1 tapering; HI shows low sensitivity.
[99]	Track monotony, strain, ACWR from s-RPE, TD, HSR across season by player position.	✓	✓	11 vs. 11 (training and matches over 10 mesocycles)	✓	✓	✗	s-RPE, TD (m), HSR (>19 km/h); calculated TM, TS, and ACWR weekly	Viper GPS (10 Hz), STATSports; CR10 scale	Wide positions had highest strain; TM and ACWR varied by position and phase.	Positional profiling reveals differences in load tolerance and workload risk markers.
[100]	Compare training intensity, as well as load and wellness indicators between starters and non-starters in elite youth football across mesocycles.	✓	✓	11 vs. 11 (training and match days across season)	✓	✓	✓ (Psychological)	s-RPE (CR10); Hooper Index (fatigue, sleep, stress, DOMS), GPS (TD, HSR, Sprint); ACWR, monotony, strain	STATSports Apex GPS (10 Hz), CR10 RPE scale, wellness questionnaires	Starters presented higher training volume and internal load; HI scores consistent across roles.	Playing status influences training volume more than internal perception; highlights the need for compensatory sessions.
[101]	Assess external training load variation across different training modes (SSG, LSG, TG) and microcycle moments.	✗	✓	SSG, LSG, TG; MD-5 to MD-1	✓	✓	✗	GPS (TD, HSR, ACC/DEC, Sprint, PlayerLoad); analysis by session type and day of week	STATSports Apex GPS (10 Hz); monitoring entire microcycle	SSG had higher ACC/DEC; LSG promoted higher sprint and HSR; TG offered moderate-intensity profile.	Exercise type affects intensity profiles; periodisation should adjust to targeted performance outcomes.
[102]	Investigate seasonal variation in training load and wellness in elite youth players (U15–U19) over 20 weeks.	✓	✓	11 vs. 11 (competitive season; U15–U19)	✓	✗	✓ (Psychological)	s-RPE (CR10); wellness index (fatigue, stress, DOMS, mood); TQR; weekly monitoring	CR10 scale, questionnaire-based HI; daily log via Excel system	U17/U19 had higher TL and fatigue; younger players showed more variable wellness scores.	Training adaptation depends on age; older players handle load better; load should be scaled to age group.
[103]	Compare external load between pre-season and in-season microcycles in elite female players.	✓	✓	11 vs. 11; M1-M4 (pre-season), M5 (in-season)	✓	✓	✗	TD, HSR (>15 km/h), ACC/DEC by zone (>1–4 m/s^2^); weekly comparisons M1–M5	PlayerTek (10 Hz) + 100 Hz accelerometer; per session per player	M3 had highest overall loads; ACC4 and DEC4 peaked in M4; only these decreased in M5.	Loads maintained throughout the season except ACC/DEC extremes; supports smooth load transition strategy.
[104]	Compare RPE-based internal load during SSG, LSG, and LSG-Sm in pro players by position.	✗	✓	SSG (4 vs. 4), LSG (10 vs. 10), LSG-Sm (9 vs. 9 small field)	✓	✓	✓ (Psychological)	RPE (CR10); comparison of intensity per game format and by playing position	CR10 RPE collected post-session; French/German/Italian translations used	Wide forwards had highest RPE in LSG-Sm; SSG > LSG in perceived load.	Game format influences perceived exertion; positional needs should guide training load adjustment.
[105]	Explore prediction of wellness index from internal TL using machine learning in sub-elite players.	✓	✓	Weekly microcycle (MD-5 to MD + 1)	✓	✗	✓ (Psychological)	s-RPE × duration; wellness index (fatigue, sleep, pain, mood, stress); Machine Learning model	CR10 RPE; wellness index (Likert scale); data input to ordinal regression model	WI predicted by TL of day before (r = 0.72); ML model outperformed random classification (39% vs. 21%)	ML can support readiness tracking from TL history; useful for training periodisation decisions.
[106]	Analyse effects of combining SSG with strength/power training on fitness in U19 players.	✗	✓	SSG-based training over 19 weeks	✓	✗	✗	RPE (CR10); GPS (distance, HSR, ACC/DEC > 2 and 4 m/s^2^); jump, sprint, agility, strength tests	FieldWiz GPS (10 Hz), RPE scale; MatLab analysis routines	Improved strength (d = 0.83), jump (d = 0.90); MD-3 exhibited highest HSR, MD-4 exhibited highest ACC/DEC.	Combined SSG + gym enhances physical fitness in youth; targeted structure is key.
[107]	Compare internal load and technical actions in different SSG formats.	✓	✓	2 vs. 2 to 5 vs. 5 SSGs	✓	✓	✓ (Psychological)	Youth players performed multiple SSG formats, with HR, RPE, and technical data recorded	Polar HR monitor, CR10 RPE, manual notation of technical actions	CR10 and HR showed different internal loads across SSG formats; more technical actions in larger formats.	Larger SSGs promote more technical and lower internal load; load varies with format.
[108]	Assess impact of minimum effort duration on measuring ACC/DEC across durations and initial velocities.	✗	✓	11 vs. 11 training (professional level)	✓	✓	✗	ACC/DEC analysed without minimum effort threshold; initial velocity range considered	Catapult Vector (10 Hz GPS + GNSS); OpenField software; position-specific analysis	Most ACC/DEC <1 s; differences in intensity between positions only visible 0.7–2.5 s	Effort duration thresholds exhibit bias in ACC/DEC analysis; accurate monitoring requires full spectrum analysis.
[35]	Quantify weekly TL and recovery status in U15, U17, and U19 sub-elite players across the microcycle.	✗	✓	11 vs. 11; MD-3, MD-2, MD-1	✓	✗	✓ (Psychological)	TD, AvSpeed, Sprint, rHSR, HMLD, ACC/DEC; s-RPE; TQR (6–20 scale)	STATSports Apex GPS (10 Hz), CR10, TQR pre-session	Older players (U17/19) had higher external TL; TQR similar across groups; MD-1 showed lowest TL.	Training load scales with age; tapering effective before match day in all age groups.
[109]	Compare weekly TL between starters and non-starters in sub-elite youth teams.	✓	✓	11 vs. 11; standard microcycle	✓	✓	✓ (Psychological)	TD, HSR, ACC/DEC, DSL; HRmax, RPE (CR10), TQR; internal/external TL across MD-3 to MD-1	STATSports Apex (18 Hz GPS), Garmin HR; Excel logs for RPE and TQR	Non-starters had higher training volume; compensatory training needed to match starters’ match load.	Weekly TL balanced when playing time adjusted; training needs differ by player role.
[12]	Apply PCA to reduce dimensionality of internal and external TL and describe resultant equations for TL monitoring in sub-elite youth football.	✗	✓	11 vs. 11 (MD-3, MD-2, MD-1; U15–U19)	✓	✓	✗	TD, HMLD, DSL, ACC/DEC; rHSR, SPR; HRmax, AvHR, %HRmax, TRIMP; TQR, sRPE, age, maturation offset; PCA model	STATSports Apex (18 Hz), Garmin HR band (1 Hz), TQR and RPE (Borg 6–20)	PCA explained 68.7% variance in 5 components; DEC, SPR, AvHR, age, MRS were main TL indicators.	Provides composite equations from TL measures; supports efficient TL tracking across microcycles using PCA-derived factors.
[110]	Compare SAQ vs. SSG effects on cognitive and physical variables in youth football	✗	✓	SAQ and SSG drills	✓	✗	✓ (Technical)	4-week intervention with pre-post sprint, COD, and cognitive performance assessment	Photocells for Sprint, as well as Flanker and Visual Search tasks	Both groups improved sprint and cognitive measures; SAQ better for physical; SSG for cognitive response.	SSG enhanced cognitive function; SAQ improved sprint and COD performance.

✓ means the study included this variable; ✗ means the study did not include it.

## Data Availability

The data supporting the findings of this study are not publicly available due to privacy or ethical restrictions but can be provided by the corresponding author upon reasonable request.

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
