# Peer review of "Shaping Training Load, Technical–Tactical Behaviour, and Well-Being in Football: A Systematic Review"

_sports, 2025, doi:10.3390/sports13080244_

Round 1
Reviewer 1 Report
Comments and Suggestions for Authors
As I engaged with this systematic review on training load, technical-tactical behavior, and well-being in football, I was struck by the sheer breadth of its ambition. The authors sought to map a fragmented field and bridge gaps between disparate performance domains. In many ways, I found the work commendable—it is comprehensive in scope, methodologically grounded in PRISMA and PICOS protocols, and attentive to issues of methodological quality. The inclusion of 46 studies spanning professional, sub-elite, and youth levels offers a valuable cross-section of current practice. Still, I found myself grappling with a tension that persists throughout the manuscript. While the dataset is rich and the literature covered is expansive, the analysis often stops short of fully synthesizing these dimensions into a coherent conceptual model. It is precisely this integration—between load, context, and well-being—that the study calls for, but does not quite deliver.
In my view, the most valuable contribution of this paper lies in its diagnostic function, which identifies the fragmentation and unevenness of the current research landscape. However, the discussion could be more ambitious in proposing how this integration might be methodologically or practically achieved. I appreciated the inclusion of machine learning examples and multidimensional load markers, but these were treated somewhat passively, rather than being used as springboards for innovation. Overall, I believe this is a strong review that needs a more assertive critical voice and a bolder articulation of future research directions.
Specific Comments
Lines 23–41 (Abstract):
The abstract summarizes the scope and findings, but its prose feels somewhat mechanical. The repeated use of metrics and abbreviations without unpacking (e.g., “TRIMP,” “HI,” “WBI”) makes it harder for non-specialist readers to grasp the central insights. I would suggest clarifying what these measures capture conceptually and adding one sentence that reflects on the broader implications of the findings, particularly the neglected integration between domains.
Lines 44–106 (Introduction):
I appreciated the effort to ground the review in the multidimensional nature of football performance; however, I felt that the rationale for including all four domains (physical, technical, tactical, and psychological) could be strengthened. While the authors reference the need for holistic monitoring, the transition from individual studies to the systematic need for integration could be more explicitly argued.
Lines 129–169 (Selection Criteria):
This section is methodologically sound and well structured. However, I was surprised by how narrow the inclusion of well-being variables became in practice. Despite the formal inclusion of psychophysiological outcomes, many of the selected studies treat these variables descriptively rather than analytically. I think a comment on the disconnect between the inclusion criteria and the actual analytic depth would be helpful.
Lines 265–301 (Physical and Psychological Dimensions):
The variability in load measurement thresholds (e.g., HSR cutoffs ranging from 15 to 25 km/h) is well-documented here and rightly problematized. However, the implications of this heterogeneity for meta-analysis or data synthesis are underexplored. I would have liked to see more reflection on how this affects the comparability of studies and what standardization might be possible or desirable.
Lines 322–330 (Multivariate Modelling):
This is one of the most promising paragraphs in the manuscript, as it highlights the potential of advanced analytics, such as PCA and machine learning. Yet, these approaches are mentioned almost in passing. I would have appreciated a more in-depth discussion of why such techniques remain underutilized and what barriers (computational, conceptual, or cultural) prevent their broader adoption.
Lines 354–363 (Discussion Opening):
The opening of the discussion recapitulates earlier findings without offering much new insight. At this point in the paper, I was hoping for a sharper critical turn—something that goes beyond describing fragmentation to theorizing it. Why is integration so complex in football science? What structural or epistemic silos sustain it?
Lines 395–415 (Tactical Dimension):
The critique here is well taken: the tactical domain suffers from inconsistent definitions, inadequate methodological standardization, and poor integration with other data streams. I strongly agree with the authors’ call for better conceptual frameworks and cross-domain linking. I would add that there may also be philosophical reasons why tactical data are more complex to codify—they are situational, fluid, and context-dependent. A brief mention of this would deepen the argument.
Lines 417–434 (Technical Dimension):
This was one of the weaker sections. While it notes that few studies contextualize technical metrics with other indicators, it does not specify how this might be done. What kinds of research designs or analytical tools could facilitate this? Could video-annotated positional data be synchronized with fatigue indices or biomechanical markers?
Lines 472–486 (Limitations and Future Directions):
I appreciated the transparency here, especially regarding the underrepresentation of sub-elite and female cohorts. However, the authors tend to list future research directions rather than offering a conceptual roadmap. I encourage them to take a stance: What would an ideal multidimensional study look like? What are the epistemological stakes of integrated monitoring?
Lines 505–556 (Practical Implications):
This section is practical and helpful, especially for coaches and applied scientists. However, some claims felt a bit too prescriptive given the earlier acknowledgment of measurement inconsistencies. For instance, recommending individual HSR thresholds makes sense, but only if those thresholds can be reliably assessed. The caveats need to be more visible.
Author Response
Dear Reviewer,
We would like to sincerely thank Reviewer 1 for their thorough, constructive, and thoughtful evaluation of our manuscript. We greatly appreciate the time and effort invested in critically reviewing our work. Your comments were instrumental in improving the manuscript’s conceptual clarity, methodological precision, and practical relevance. Below, we provide detailed point-by-point responses to each of your comments, accompanied by explanations of the corresponding revisions. All changes have been implemented in the revised manuscript and are visible in tracked changes.
Please see the attached cover letter with peer review responses.
With best regards,
José Eduardo Teixeira.

Reviewer 2 Report
Comments and Suggestions for Authors
GENERAL COMMENTS
Major Weaknesses
- Methodological Rigor Needs Strengthening: The systematic review exhibits several critical methodological flaws. The database coverage appears insufficient, inclusion criteria are unclear, and the quality assessment thresholds seem arbitrary. These issues collectively undermine the reliability of the findings.
- Lack of Genuine Integration: Despite the title and objectives promising multidimensional integration, the review largely presents findings from different performance domains in isolation. A true synthesis of interdisciplinary insights is largely absent.
- Limited Novelty in Conclusions: The conclusions primarily confirm existing knowledge within football performance monitoring. There's a missed opportunity to advance theoretical understanding or propose novel practical frameworks.
- Insufficient Data Synthesis: The presentation of data is largely descriptive and lacks analytical depth. Opportunities for appropriate meta-analytical approaches, especially where quantitative synthesis would be beneficial, were not pursued.
- Inadequate Practical Applications: The review struggles to translate its findings into actionable monitoring frameworks that practitioners could readily use, thereby limiting its practical value.
Minor Weaknesses
- Presentation Clarity: The tables, particularly Tables 5 and 6, are overly complex and difficult to interpret, often containing redundant information.
- Inconsistent Terminology: The inconsistent use of terms across different performance dimensions detracts from the clarity and precision of the review.
- Limited Critical Discussion: The discussion section could benefit from a more thorough critical evaluation of study limitations and any contradictory findings.
SPECIFIC COMMENTS
Abstract
Lines 25-26, Page 1: The phrase "limited evidence on integrated monitoring in youth and sub-elite contexts" needs further quantification. It would be helpful to define "how limited" and what constitutes "sufficient evidence" in this context.
Line 35, Page 1: The finding "Only five studies used multivariate models" is a significant limitation and should be prominently highlighted as such, rather than being less emphasized within the results section.
Introduction
Lines 85-88, Page 2: The assertion that previous reviews focused "largely on physical and psychological variables while often neglecting tactical and technical components" requires robust citation support.
Lines 98-107, Page 3: The stated hypothesis lacks specificity and testable elements. Clarification is needed on what precisely constitutes "interactions" between dimensions.
Methods
Search Strategy:
Lines 115-116, Page 3: Limiting the search to only PubMed, Scopus, and Web of Science is insufficient for a truly comprehensive systematic review. Major databases such as EMBASE and SPORTDiscus are conspicuously absent.
Table 1, Page 3: The Boolean search strategy appears overly complex and potentially restrictive. Its nested structure may inadvertently exclude relevant studies.
Selection Criteria:
Lines 142-148, Page 4: The inclusion criteria are excessively broad and ambiguous. The phrase "concurrently reported" variables, for instance, lacks a clear operational definition.
Lines 154-157, Page 4: The thresholds for MQCOM score (≥12) and PEDro score (≥6) are not adequately justified with appropriate references.
Quality Assessment:
Lines 171-176, Page 4: The use of different quality assessment tools for different study designs introduces inconsistency in the evaluation standards, which is problematic.
STATISTICAL AND METHODOLOGICAL CONCERNS
- Missing Meta-Analysis: Despite the presence of quantitative data from multiple studies (e.g., training load metrics), no attempts were made at statistical synthesis, such as a meta-analysis.
- Heterogeneity Assessment: No formal assessment of study heterogeneity was conducted, even though obvious methodological differences across studies are apparent.
- Risk of Bias: While individual study bias assessment is mentioned, the results are neither systematically presented nor adequately discussed.
- Publication Bias: No assessment of publication bias was conducted, despite this being standard practice for systematic reviews.
Results
Study Selection:
Lines 220-228, Page 7: The PRISMA flow diagram in Figure 1 is missing its reference, making it difficult to properly evaluate the study selection process.
Sample Characteristics:
Lines 232-243, Page 8: The extremely broad age range (13.2-28.7 years) could introduce significant heterogeneity issues that have not been adequately addressed in the review.
Main Findings:
Lines 267-275, Page 10: The acknowledged inconsistency in terminology (e.g., HSR vs. HIR vs. HID) remains unresolved, which fundamentally undermines comparative analysis.
Lines 285-301, Page 11: The analysis of the psychological dimension conflates subjective measures with objective psychophysiological indicators without a clear and necessary distinction.
Tables
Table 5, Pages 13-21: This table is excessively long and contains a great deal of redundant information. Many entries could be consolidated or more appropriately moved to supplementary material.
Table 6, Pages 22-43: The table detailing methodological approaches lacks clear organization and also contains duplicate information across studies.
Discussion
Lines 358-363, Page 44: The statement about an "anticipated gap in integrated assessment" suggests a degree of confirmation bias rather than an objective analysis of the literature.
Lines 377-394, Page 44: The discussion concerning the physical dimension fails to adequately address the fundamental problem of threshold heterogeneity, which significantly undermines any comparative conclusions drawn.
Lines 454-468, Page 45: While the association section acknowledges the limited use of multivariate approaches, it doesn't adequately critique this major methodological limitation within the broader context of the review.
Limitations
Lines 472-486, Page 46: The limitations section is notably brief, especially considering the extensive methodological issues identified throughout the review. The variability in measurement techniques, in particular, warrants more critical analysis.
Conclusions
Lines 495-504, Page 47: The recommendations for future research are generic and lack the specificity needed to genuinely advance the field.
Lines 506-544, Page 47: The practical recommendations section appears largely disconnected from the actual findings presented in the results.
Author Response
Dear Reviewer,
We would like to express our sincere gratitude to Reviewer 2 for the exceptionally detailed, rigorous, and thoughtful evaluation of our manuscript. Your comments were not only intellectually stimulating but also instrumental in refining the scientific quality, conceptual clarity, and practical relevance of our work. The depth and precision of your critique challenged us to rethink and improve nearly every section of the manuscript, from methodological transparency to the integration of findings. We have thoroughly addressed each of your comments below and implemented all corresponding changes in the revised manuscript. Wherever appropriate, textual modifications have been clearly marked and justified.
Please see the attached cover letter with peer review responses.
With my best regards,
José Eduardo Teixeira.

Round 2
Reviewer 2 Report
Comments and Suggestions for Authors
PEER REVIEW REPORT
GENERAL COMMENTS
Thanks to the authors for their diligent work in revising this manuscript. I appreciate the effort put into addressing my previous feedback, which has substantially improved the paper's scientific rigor and practical relevance. While the revised manuscript is strong, I believe a few minor editorial adjustments could further enhance its conciseness and sharpen its practical recommendations before it's ready for publication.
A key observation is that despite highlighting fragmentation as a central issue, the review itself still presents some of its analyses in a somewhat fragmented manner, rather than always pushing for more integrated frameworks. Moreover, while it effectively identifies problems, the paper could offer more concrete, actionable solutions for practitioners looking to implement multidimensional monitoring strategies.
Some sections also suffer from verbose writing, with certain concepts being repeated. Streamlining these areas would greatly improve readability. Finally, I noticed a limited discussion of the real-world barriers that often prevent the adoption of integrated monitoring approaches. Expanding on this would add significant value.
SPECIFIC COMMENTS
Abstract
Line 33, Page 1: "TRIMP" should be introduced as "Training Impulse (TRIMP)" upon its first appearance.
Line 38, Page 1: For readers less familiar with the specifics, it might be helpful to briefly explain that "TRACAB®" refers to a tracking system.
Introduction
Lines 91-93, Page 2: The phrasing here feels a bit redundant. Perhaps consolidate it to something like: "Previous systematic reviews have primarily examined performance domains in isolation, often prioritizing physical and psychological variables while overlooking tactical and technical aspects."
Lines 105-119, Page 3: This paragraph is quite lengthy and contains some repetitive ideas. Breaking it into two or three more focused paragraphs would improve clarity.
Methods
Lines 138-140, Page 3: The statement that wellbeing variables were addressed "in a purely descriptive manner" seems to contradict their inclusion criteria. Please clarify this apparent inconsistency.
Line 169, Page 4: Ensure the justification for the "MQCOM score < 12" cutoff is explicitly stated and that this criterion is applied consistently throughout the study.
Results
Lines 304-308, Page 11: The text refers to "Table 5" for a summary of measures, thresholds, and metric formulas, but Table 5 isn't visible in the document. Please verify table numbering.
Lines 353-357, Page 12: The assertion that "Although promising, these approaches remain rare within the literature" would benefit from supporting quantitative data to underscore its claim.
Discussion
Lines 386-393, Page 14: The insight that fragmentation is "not merely methodological" but potentially indicative of "structural silos within football science" is compelling. However, this claim needs stronger evidence or further elaboration to fully support it.
Lines 436-441, Page 15: The discussion on heterogeneity here appears somewhat repetitive given earlier content. Consider consolidating or removing these points to avoid redundancy.
Lines 531-537, Page 17: The section on "methodological inertia" makes some strong claims that could benefit from a more nuanced discussion, perhaps exploring potential contributing factors or counterarguments.
Technical Issues
Line 116, Page 3: Please ensure consistent formatting for abbreviations like "e.g., HI, TQR, Likert-type scales."
Page 6, Table 2: Several entries list "ND" (Not Described). It would be beneficial to discuss this limitation within the main text, acknowledging where information was incomplete.
Lines 573-582, Page 18: The recommendations provided are comprehensive but could be more impactful if they were prioritized for practical implementation. Perhaps categorize them or highlight the most critical ones.
Author Response
Dear Reviewer,
We would like to thank you very much for the continued critical and constructive feedback. We greatly appreciate your recognition of the improvements made and your suggestions for further refinement. In this second round of revisions, we have carefully considered all additional comments and implemented the corresponding changes in the manuscript. Where relevant, modifications are highlighted in yellow in the revised document.
Please, see the attached cover letter with point-by-point responses.
With my best regards,
José Eduardo Teixeira
